# Killer-like receptors and GPR56 progressive expression defines cytokine production of human CD4$^+$ memory T cells

Kim-Long Truong[1,7], Stephan Schlickeiser[1,2,7], Katrin Vogt[1], David Boës[1], Katarina Stanko[1], Christine Appelt[1], Mathias Streitz[1], Gerald Grütz[1,2], Nadja Stobutzki[1], Christian Meisel[1], Christina Iwert[1], Stefan Tomiuk[3], Julia K. Polansky[2,4], Andreas Pascher[5], Nina Babel[2,6], Ulrik Stervbo [6], Igor Sauer [5], Undine Gerlach[5] & Birgit Sawitzki[1,2]

All memory T cells mount an accelerated response on antigen reencounter, but significant functional heterogeneity is present within the respective memory T-cell subsets as defined by CCR7 and CD45RA expression, thereby warranting further stratification. Here we show that several surface markers, including KLRB1, KLRG1, GPR56, and KLRF1, help define low, high, or exhausted cytokine producers within human peripheral and intrahepatic CD4$^+$ memory T-cell populations. Highest simultaneous production of TNF and IFN-γ is observed in KLRB1$^+$KLRG1$^+$GPR56$^+$ CD4 T cells. By contrast, KLRF1 expression is associated with T-cell exhaustion and reduced TNF/IFN-γ production. Lastly, TCRβ repertoire analysis and in vitro differentiation support a regulated, progressive expression for these markers during CD4$^+$ memory T-cell differentiation. Our results thus help refine the classification of human memory T cells to provide insights on inflammatory disease progression and immunotherapy development.

[1] Institute of Medical Immunology, Charité – Universitätsmedizin Berlin, Freie Universität Berlin, Humboldt-Universität zu Berlin and Berlin Institute of Health, 13353 Berlin, Germany. [2] Berlin-Brandenburg Center for Regenerative Therapies (BCRT), Charité – Universitätsmedizin Berlin, 13353 Berlin, Germany. [3] Milteny Biotec GmbH, 51429 Bergisch Gladbach, Germany. [4] German Rheumatism Research Centre, 10117 Berlin, Germany. [5] Department of Surgery, Charité – Universitätsmedizin Berlin, 13353 Berlin, Germany. [6] Medical Clinic I, Marien Hospital Herne, University Clinic of Ruhr-University Bochum, 44625 Herne, Germany. [7] These authors contributed equally: Kim-Long Truong, Stephan Schlickeiser. Correspondence and requests for materials should be addressed to B.S. (email: birgit.sawitzki@charite.de)

CD4+ helper T cells coordinate the immune response against invading pathogens and malignancies[1–5]. However, they also play a pathological role in the development of various inflammatory and/or autoimmune disorders[6,7].

In association to their differentiation state, CD4+ T-cell populations vary in their migratory behavior, cytokine production, and proliferative capacity, as well as effector function[8,9]. Naive T cells ($T_N$) upon primary antigen encounter in secondary lymphoid organs exert only delayed effector functions[10,11]. In contrast, memory T cells show an accelerated and intensified response to antigen re-encounter resulting in rapid antigen clearance. However, the functional repertoire of memory T cells is manifold and subpopulations vary in their location, protection capacity, and longevity[10,11]. Therefore, researchers have urged to identify phenotypic properties that help to distinguish memory T-cell subpopulations[12,13]. Based on the expression patterns of lymph node homing receptors (CD62L or CCR7) and CD45 splice variants, CD8+ and CD4+ T cells have been classified into CD45RA+CCR7+ naive ($T_N$), CD45RA−CCR7+ central memory ($T_{CM}$), CD45RA−CCR7− effector memory ($T_{EM}$), and CD45RA+CCR7− terminally differentiated effector memory ($T_{EMRA}$) cells[14–16]. $T_{CM}$ being CCR7+, such as $T_N$ cells, circulate between blood and lymphoid compartments, and have a high proliferative and self-renewal capacity. Both CCR7− $T_{EM}$ and $T_{EMRA}$ subsets are excluded from lymphatic organs and migrate via the blood to peripheral tissue[15,16] where they elicit an immediate cytokine-driven cytotoxic immune response against re-occurring infections[9,17]. Although $T_{EM}$ cells show generally high cytokine production potential, contrasting findings have been published for $T_{EMRA}$ cells, reporting either high or low cytokine production potential; the latter being attributed to an exhausted state with characteristics of end-stage differentiation, showing acquirement of killer cell lectin-like G1 (KLRG1) and CD57 but loss of CD28 and CD27 expression[9,18–21]. Similar findings suggesting multifunctionality within a subset have also been observed for $T_{EM}$ cells[21,22]. Thus, $T_{EM}$ and $T_{EMRA}$ populations defined by the CD45RA/CCR7 classification seem to represent a pool of cells, which are functionally and phenotypically heterogeneous. This is particularly true for CD4+ memory T cells.

Due to their high pro-inflammatory cytokine production potential, CD4+ memory T cells are key promoters of chronic inflammation when physiological regulatory circuits fail. Therefore, it is not surprising that increased proportions and absolute numbers of $T_{EM}$ and also $T_{EMRA}$ cells have been observed in patients suffering from chronic inflammatory diseases[23,24]. However, due to high intra-subset heterogeneity and possible partial functional overlap of $T_{EM}$ and $T_{EMRA}$, it seems likely to be that T cells with high cytokine secretion properties might be driving chronic inflammation irrespective of their CD45RA/CCR7 phenotype.

Therefore, in this study we aim for further characterization of functional heterogeneity of human CD4+ T-cell subsets at the single-cell level, including the identification of reliable surface markers correlating with their cytokine production properties. Here we perform bulk and single cell gene expression profiling of purified human CD4+ $T_N$, $T_{CM}$, $T_{EM}$, and $T_{EMRA}$ cells from the peripheral blood of healthy individuals and identify different combinations of the surface markers KLRB1, KLRG1, GPR56, and KLRF1 to be suitable to describe the development of human CD4+ memory T cells with varying cytokine production potential. Expression of KLRB1, KLRG1, and GPR56 is associated with high tumor necrosis factor (TNF)/interferon (IFN)-γ co-expression potential, whereas acquisition of KLRF1 expression during terminal differentiation results in a reduction of the cytokine production capacity. This KLR and GPR56-based classification allows for a more precise definition of functional states of TNF/IFN-γ producers as compared to the classical $T_{EM}$ or $T_{EMRA}$ gating, respectively. Importantly, this correlation between marker expression and cytokine production is also validated in blood and intra-tissue CD4+ T cells from patients with inflammatory liver diseases. Our data thus support a human CD4+ memory T-cell classification scheme that is more precise than the CD45RA/CCR7-based categorization.

## Results

**Overlapping gene signatures of CD4+ $T_{EM}$ and $T_{EMRA}$ cells.** To gain more insights into common vs. different phenotypic and functional properties of CD4+ $T_{EM}$ and particularly $T_{EMRA}$ cells, we performed comparative gene expression profiling of sorted CD4+ $T_N$, $T_{CM}$, $T_{EM}$, and $T_{EMRA}$ cells from the peripheral blood of healthy individuals. We focussed on genes encoding cell surface proteins, in order to identify phenotypic markers associated with distinct functional properties (e.g., cytokine production potential) of $T_{EM}$ and especially $T_{EMRA}$ cells. Therefore, we performed an intersection analysis as explained in Methods between $T_{EMRA}$/$T_{EM}$ vs. $T_N$, or $T_{EMRA}$/$T_{EM}$ vs. $T_{CM}$ cells (Fig. 1a), and $T_{EMRA}$ vs. $T_N$, $T_{EMRA}$ vs. $T_{CM}$, or $T_{EMRA}$ vs. $T_{EM}$ (Fig. 1b), to identify genes that were highest expressed in $T_{EMRA}$ cells (Fig. 1b) but also significantly increased in both CD4+ $T_{EM}$ and $T_{EMRA}$ cells (Fig. 1a) compared with CD4+ $T_N$/$T_{CM}$ or with $T_N$ cells only (see also Supplementary Table 1). Among the top-ranked genes upregulated in both CD4+ $T_{EM}$ and $T_{EMRA}$ cells were genes previously shown to be highly expressed in more differentiated T cells (e.g., *FASLG*) as well as genes encoding various Toll-like receptors or *HLA-DR beta* chains, but especially genes associated with natural killer (NK) cells. Among those NK cell genes were members of the killer-like receptor family (e.g., *KLRG1* and *KLRF1*, *NKG7*, *CD300A*, and *CD300C*). In addition, genes encoding proteins regulating cell migration and adhesion such as *S1PR5*, *CX3CR1*, and *ADGRG1* (also known as *GPR56*) were highly upregulated. Furthermore, the microarray analysis identified genes whose expression was specifically increased in $T_{EM}$ cells, such as genes encoding c-Kit or the killer-like receptor *KLRB1* (see also Supplementary Table 1). Taken together, we could identify a set of genes encoding surface markers, which were significantly higher expressed in more differentiated human CD4+ $T_{EM}$ and $T_{EMRA}$ cells.

**Heterogeneity in gene expression of $T_{EMRA}$ and $T_{EM}$ cells.** We used the identified CD4+ $T_{EM}$− and $T_{EMRA}$-specific genes to investigate whether their expression was homogeneous or could be attributed to certain cell subsets within CD4+ $T_{EM}$ and $T_{EMRA}$ cells. For this, we applied single-cell separation combined with gene candidate-specific quantitative reverse-transcriptase PCR (qRT-PCR) in separated CD4+ $T_N$, $T_{CM}$, $T_{EM}$, and $T_{EMRA}$ cells, as well as CD4+CD25highCD127low regulatory T cells (Treg) from healthy individuals. In addition to the expression of our identified gene candidates, we also analyzed expression of *TBX21*, *GATA3*, *EOMES*, *RORC*, *FOXP3*, *SELL*, and *CD45RA* (see Supplementary Table 2 for complete gene list). We performed an unsupervised hierarchical cluster analysis of all genes giving a signal in at least ten cells and showing no cross-reactivity with genomic DNA.

Nearly all of the $T_{EMRA}$ and the majority of the $T_{EM}$ cells clustered separately from the other T-cell subsets, which was in part due to the inclusion of lineage-specific genes such as *EOMES* (Fig. 1c). Also, as expected, Treg cells clustered separately in a highly homogeneous cluster, underlining the clear separation of this immunosuppressive subset from all other pro-inflammatory subsets. A high proportion of $T_{EM}$ but also $T_{EMRA}$ cells transcribed *EOMES* and also *GATA3*, whereas *RORC* was only

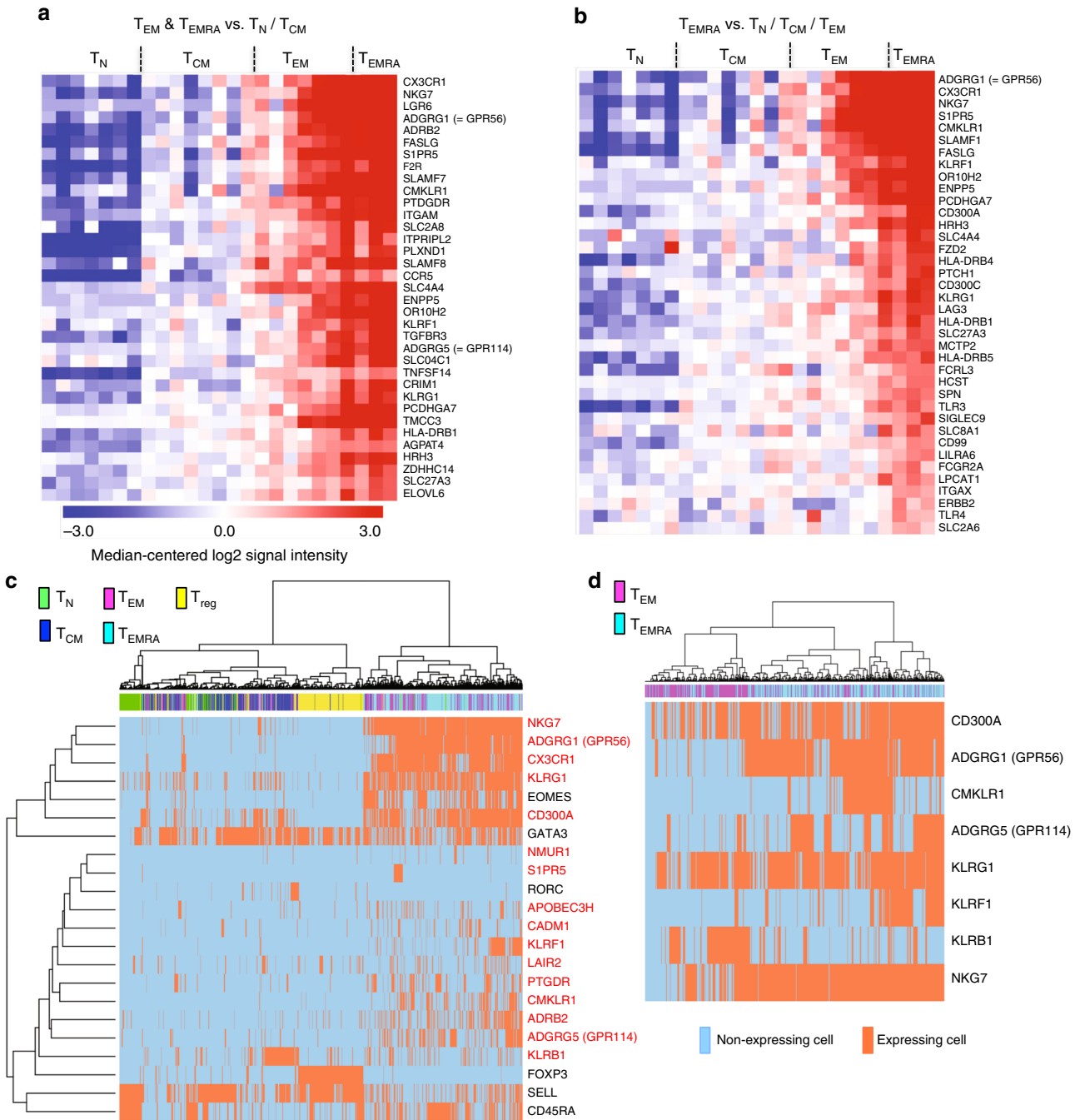

**Fig. 1** Specific mRNA expression profile and heterogeneity of CD4+ TEM and TEMRA cells. **a**, **b** Heatmaps of genes with significantly increased expression in **a** $T_{EM}/T_{EMRA}$ compared to $T_N/T_{CM}$ cells and **b** $T_{EMRA}$ compared with $T_N/T_{CM}/T_{EM}$ cells identified by an intersection analysis. Gene expression was determined in human CD4+ $T_N$, $T_{CM}$, $T_{EM}$, and $T_{EMRA}$ cells sorted from peripheral blood of healthy subjects ($n = 3$–8, each consisting of 1–3 pooled sorted samples). The scale of both heatmaps is identical. **c** Single-cell profiling of $T_{EM}$- and $T_{EMRA}$-specific gene expression shown as unsupervised hierarchical cluster analysis of 16 gene candidates (red) and additional genes (black) in single blood CD4+ $T_{EMRA}$ ($n = 226$), $T_{EM}$ ($n = 199$), $T_{CM}$ ($n = 186$), $T_N$ ($n = 94$), and Treg cells ($n = 178$) from four healthy donors. **d** Unsupervised hierarchical cluster analysis of single-cell gene expression results from identified natural killer cell-associated markers in blood CD4+ $T_{EM}$ ($n = 260$) and $T_{EMRA}$ ($n = 276$) cells of five healthy individuals. Classification as expressing and non-expressing cells based on individually defined limit of detection (LoD) Ct values. Data are provided with the Source Data file

expressed by a minor fraction of $T_{CM}$, $T_{EM}$, and Treg cells but not by $T_{EMRA}$ cells (Supplementary Table 3).

Single-cell quantitative PCR (qPCR) analysis not only validated the selective expression pattern of most of the gene candidates observed in bulk analysis but also revealed that only a few CD4+ $T_{EM}$- and $T_{EMRA}$-specific genes such as *NKG7*, *GPR56*, or *KLRG1* were expressed by nearly all $T_{EM}$ and $T_{EMRA}$ cells (Fig. 1c, d and Supplementary Table 3). The majority of genes showed a more

heterogeneous expression pattern with only a fraction (e.g., *KLRB1*, *KLRF1*, *CMKLR1*, and *ADRB2*) or sometimes even a minority of CD4+ $T_{EM}$ or $T_{EMRA}$ cells transcribing the genes (e.g., *S1PR5*, *CADM1*). The variability in NK cell-associated marker expression (Fig. 1d) was especially apparent.

As we were particularly interested in understanding heterogeneity associated with variations in functionality and differentiation status of the T cells, we investigated the heterogeneously

expressed genes in more detail at protein level. We concentrated on KLRB1, KLRG1, GPR56, and KLRF1 as we wanted to include markers with homogeneous and those with heterogeneous expression as well. We achieved reproducible high-resolution antibody staining in flow cytometry.

We detected a progressive increase in expression of all four markers starting from $T_N$ to $T_{EMRA}$ cells (Fig. 2a) with upregulation of KLRB1 at an early memory stage (central memory stage). In contrast, expression of the other three markers increased later during memory/effector cell development: KLRG1 was expressed by ~50% of the $T_{EM}$ and nearly all $T_{EMRA}$ cells, whereas GPR56 and especially KLRF1 were particularly upregulated in even later stages of differentiation with only $T_{EMRA}$ cells displaying a relatively high expression of these markers (Fig. 2a).

Thus, we could validate the selective expression pattern of killer-like receptors and GPR56 in $T_{EM}$ and $T_{EMRA}$ cells. In concordance with the single-cell mRNA expression analysis, the expression patterns were still heterogeneous and not all of the $T_{EM}$ and $T_{EMRA}$ cells stained positive for these four markers.

In addition, we studied expression of co-inhibitory receptors known to become upregulated during T-cell differentiation or T-cell dysfunction/exhaustion such as PD-1 and TIGIT. Of interest, expression of both receptors was highest in $T_{EM}$ or $T_{CM}$ cells, respectively, but reduced in $T_{EMRA}$ cells (Fig. 2a). Our analysis of other co-inhibitory receptors such as TIM-3 and LAG-3 revealed rather high surface expression on $T_N$ and $T_{CM}$, and low expression on $T_{EM}$ and $T_{EMRA}$ cells (Supplementary Fig. 1A), whereas, in contrast, mRNA expression of LAG-3 was only upregulated in $T_{EMRA}$ cells and, to a lesser degree, $T_{EM}$ cells (Fig. 1b). Supplementary Table 4 summarizes the expression pattern of previously defined and our proposed surface markers on human and murine CD4[+] and CD8[+] T cells that distinguish between activated, memory, and dysfunctional/exhausted states.

When comparing single-cell expression data on mRNA to respective data on protein level, we observed nearly identical frequencies of GPR56- and KLRG1-expressing cells (Fig. 2b). In contrast, proportion of KLRF1 and especially KLRB1-positive cells varied significantly for $T_{EMRA}$ cells pointing to fluctuations in gene transcription.

**KLR and GPR56 expression-associated cytokine production.** Next, we tested a potential correlation between cytokine production of CD4[+] T cells and surface marker expression of KLRB1, KLRG1, GPR56, and KLRF1 each. For this, peripheral blood mononuclear cells (PBMCs) of healthy individuals were short-term stimulated using phorbol 12-myristate 13-acetate (PMA)/Ionomycin, followed by staining of the respective surface markers and intracellular cytokines, namely TNF, IFN-γ, interleukin (IL)-4, and IL-17. We observed a general trend of KLRB1[+], KLRG1[+], and GPR56[+] T cells displaying high potential to produce cytokines, with especially the KLRG1[+] group of cells containing a particular high frequency of TNF and IFN-γ producers (Fig. 3a). In contrast, IL-4 production showed no association with expression of KLRs and IL-17A production was mostly observed within KLRB1[+] cells (Supplementary Fig. 2A) as previously described[25]. With single KLRB1-expressing CD4[+] T cells mainly residing within $T_{CM}$ cells, we detected only IL-17A expression in $T_{CM}$ but not in $T_{EM}$ or $T_{EMRA}$ cells (Supplementary Fig. 2B). However, total proportion of IL-4 or IL-17A producers was rather low within peripheral CD4[+] T cells of healthy controls.

In order to visualize defined co-expression patterns of the surface markers associated with TNF and IFN-γ production, we created t-distributed stochastic neighbor embedding (t-SNE) maps arranging all conventional CD4[+] T cells (pre-gated on CD3[+]CD4[+] cells with exclusion of CD25[high]CD127[low] Treg

cells) according to their similarity in surface marker (CCR7, CD45RA, KLRB1, KLRG1, GPR56, and KLRF1) and cytokine expression (TNF, IFN-γ, IL-4, and IL-17A; Fig. 3b). The two-dimensional shape of all t-SNE plots is based on the overall similarities between the acquired conventional CD4[+] T cells. The color code of each t-SNE plot reflects the corresponding distribution of marker-positive and -negative T cells, which, e.g., allows the identification of CCR7-negative, CD45RA-negative, or -positive $T_{EM}$ and $T_{EMRA}$ cells respectively, in the upper left t-SNE plots (encircled black area). TNF and IFN-γ production is common but clearly heterogeneous within the $T_{EM}$/$T_{EMRA}$ area, with certain subtypes being completely devoid of cytokine expression (blue arrows). In accordance with the single-cell gene expression results, we detected a homogeneous KLRG1 expression in nearly all cells within the $T_{EM}$/$T_{EMRA}$ area, whereas expression of the other marker was very heterogeneous.

Surprisingly, most cells in this cytokine-low area (blue arrows) express all four surface markers with KLRF1 displaying an almost exclusive expression for the $T_{EM}$/$T_{EMRA}$ subset. Furthermore, areas of high cytokine production (TNF[+] and IFN-γ[+]) contain cells that either co-express KLRB1 and KLRG1 (pink arrows) or KLRG1 and GPR56 (purple arrows). These results from visual inspection of the t-SNE maps indicated that different combinations of these surface markers are characteristic for different functional states. As the acquisition or loss of cytokine expression potential is generally linked to the differentiation state of T cells, we wanted to analyze how the expression of the aforementioned set of surface markers correlates to the differentiation pathway of memory T cells according to the CD45RA/CCR7-based classification. For this, we applied the recently described wanderlust algorithm to construct a trajectory for CD4[+] T-cell differentiation based on the classical surface marker CD45RA and CCR7, our surface marker set (KLRG1, KLRB1, KLRF1, and GPR56) and cytokine expression (TNF and IFN-γ)[26]. CD45RA[+]CCR7[+] ($T_N$) cells were defined as initiator cells, thus determining the start point of the wanderlust plot.

We then examined the relative expression pattern of our selected surface markers but also levels of intracellular TNF and IFN-γ along the developmental trajectory by plotting them against the wanderlust axis (Fig. 3c). According to this analysis, KLRB1 expression was the first marker to be acquired during CD4[+] memory T-cell differentiation, a result which is confirmed by our bulk and single-cell-based gene expression analyses (Figs. 1 and 2). Subsequently, cells started to upregulate KLRG1 followed by a nearly parallel induction of GPR56. KLRF1 expression was only acquired at a late stage during memory T-cell differentiation. Interestingly, simultaneous to the upregulation of KLRB1, T cells obtained the potential to produce TNF and, with a slight delay, also IFN-γ. Whereas KLRB1 and KLRG1 showed a nearly constant increase in expression during differentiation, GPR56 and KLRF1 expression followed a two-phase pattern. Late-stage differentiated CD45RA re-expressing CD4[+] T cells acquired very high KLRG1, GPR56, and KLRF1 expression but a reduction in KLRB1 expression concurrent with a decline in TNF and IFN-γ production.

**Combinations of KLRs and GPR56 define memory T-cell states.** The wanderlust analysis revealed a progressive acquisition of our surface markers during memory T-cell differentiation in the following order: KLRB1, KLRG1, GPR56, and KLRF1. Based on this, we analyzed whether T-cell subsets defined by this scheme would indeed recapitulate or even refine the known correlation between cytokine expression potential and stages of differentiation, which is currently describing $T_N$ cells as low, $T_{EM}$

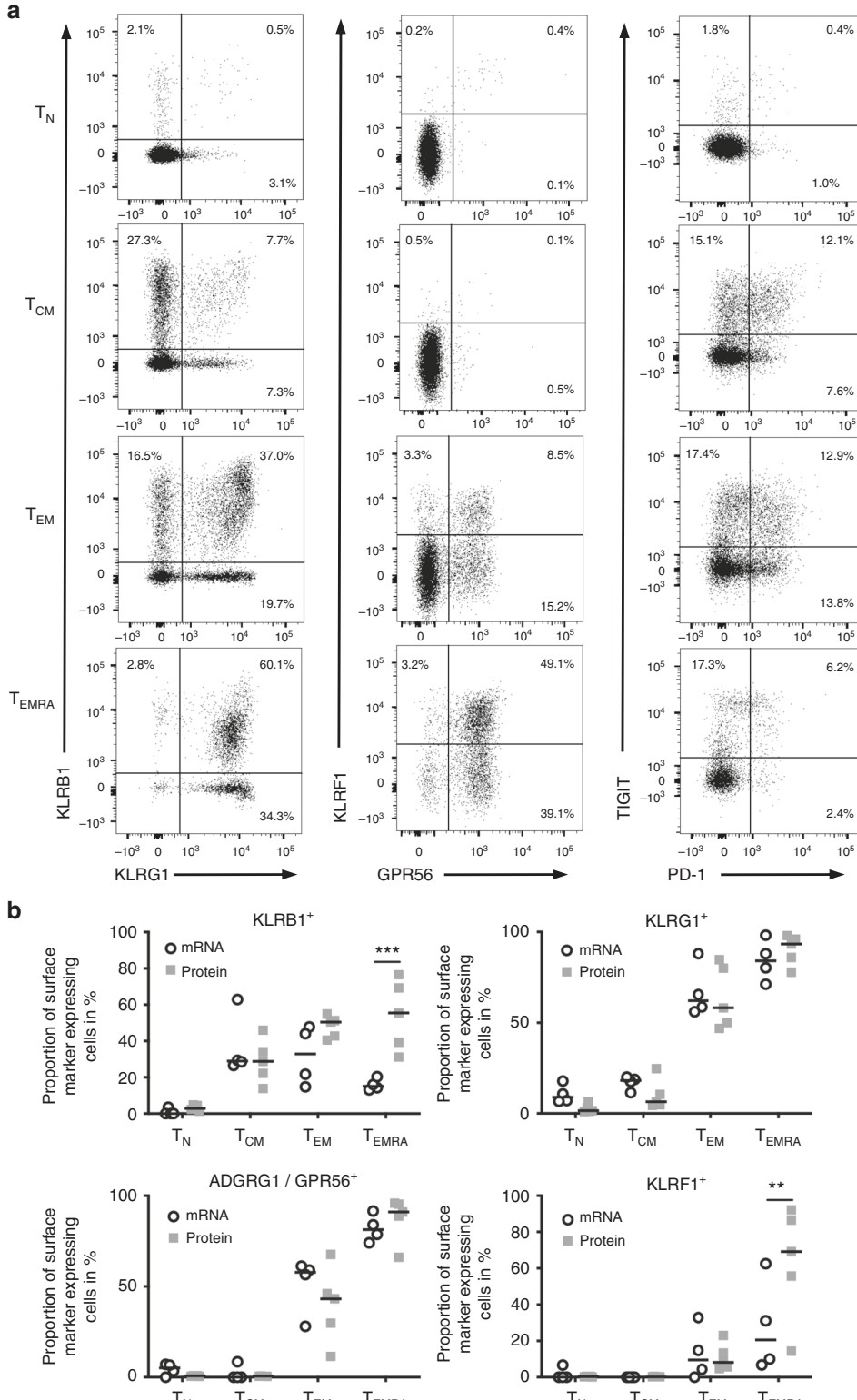

**Fig. 2** Heterogeneous surface expression of killer-like receptors and GPR56 in CD4+ T cells. **a** Exemplary dot plots of KLRB1, KLRG1, KLRF1, GPR56, TIGIT, and PD-1 surface expression in gated CD4+ $T_N$, $T_{CM}$, $T_{EM}$, and $T_{EMRA}$ cells from blood of healthy donors. **b** Comparison of KLRB1-, KLRG1-, KLRF1-, and GPR56-positive cell frequencies obtained either from single-cell gene expression analysis (mRNA, $n = 209$ ($T_N$), 260 ($T_{CM}$), 396 ($T_{EM}$), and 248 ($T_{EMRA}$) from four donors) or flow cytometric analysis (protein, $n = 5$) within CD4+ $T_N$, $T_{CM}$, $T_{EM}$, and $T_{EMRA}$ cells. Data are shown as individual scatter plots with median. Statistical analysis by two-way ANOVA and Sidak's multiple comparison test. $**p < 0.01$, $***p < 0.001$

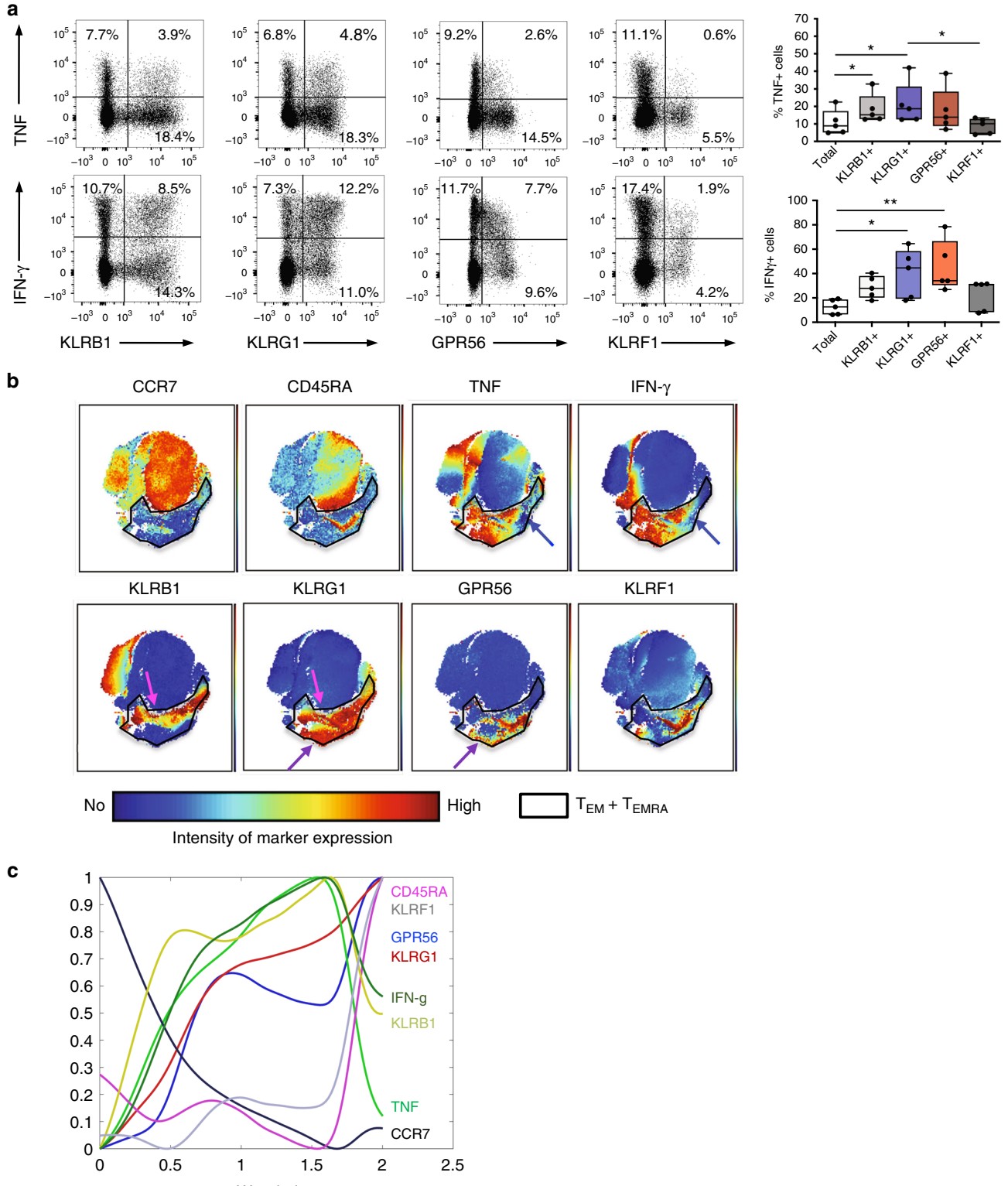

**Fig. 3** Progressive expression of KLRs and GPR56 is associated with cytokine production during CD4+ memory T-cell development. **a** Association of surface marker expression with cytokine expression in CD4+ T cells from peripheral blood of healthy individuals shown as exemplary dot plots and summarizing box and whisker plots (% of cytokine expressing cells within total or marker positive subset, whiskers extend to the minimum and maximum) of $n = 5$. Cells were restimulated with PMA and Ionomycin. **b** Representative t-SNE plots showing surface marker and cytokine expression pattern of pre-gated CD4+ T cells (excluding CD25highCD127low Treg cells) upon short-term phorbol 12-myristate 13-acetate (PMA)/Ionomycin stimulation. The highlighted area marks CD4+ $T_{EM}$ and $T_{EMRA}$ cells identified by the absence of CCR7 and the expression pattern of CD45RA. **c** Wanderlust analysis based on the trajectory of CD45RA and CCR7. Relative median surface marker and intracellular cytokine expression within CD4+ T cells from blood of five healthy individuals upon short-term PMA/Ionomycin stimulation are shown as described within Methods. P-values were determined by non-parametric matched-pairs Friedman's test with post-hoc Dunn's multiple comparison test. *$p < 0.05$, **$p < 0.01$

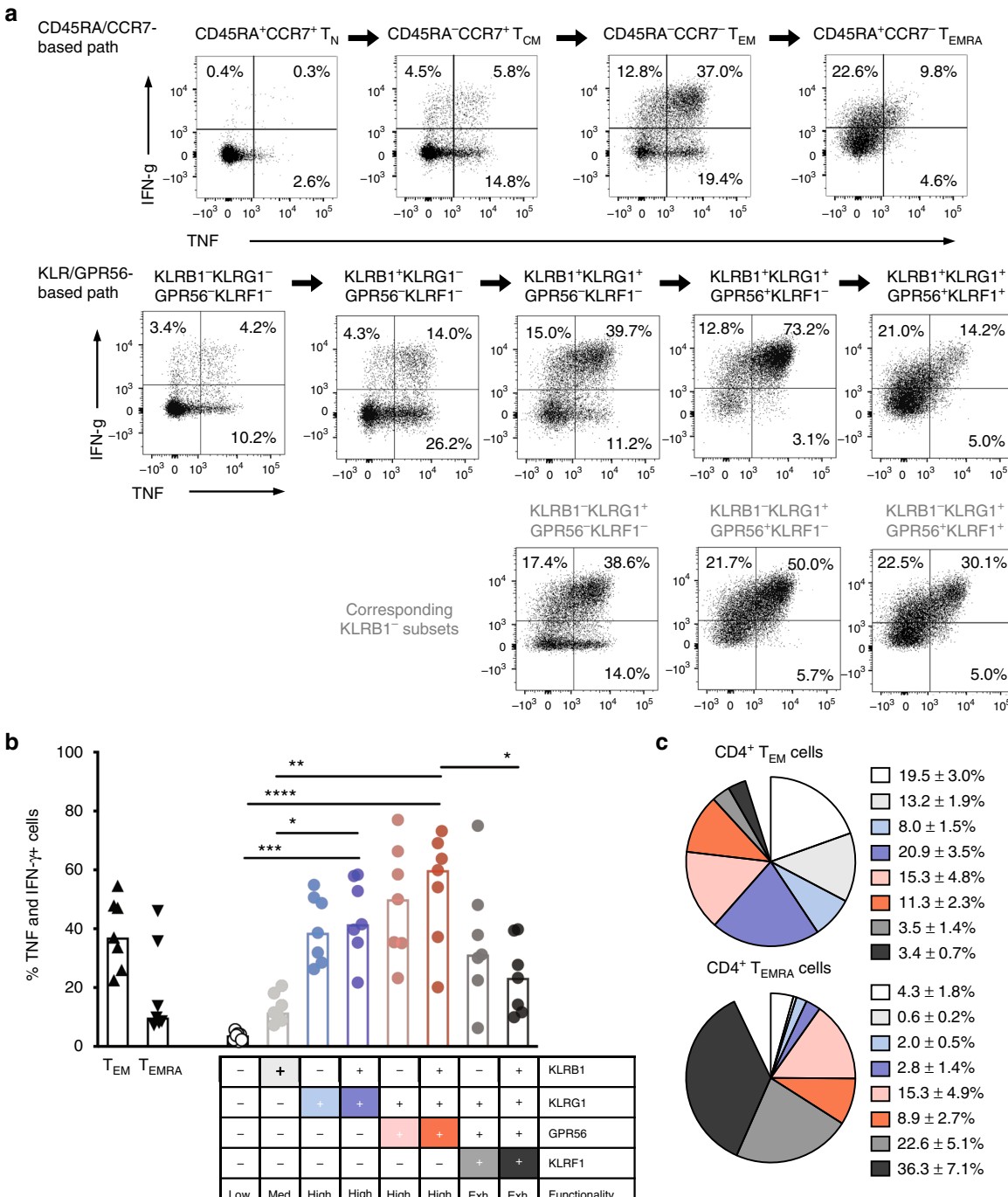

**Fig. 4** Combinational expression of KLR's and GPR56 recapitulates human CD4+ memory T-cell development. **a** Exemplary dot plots of TNF and IFN-γ production upon short-term PMA/Ionomycin stimulation within conventionally gated (CD45RA/CCR7-based path) or KLR/GPR56-based gated CD4+ memory T cells. The plots are arranged according to the anticipated developmental pathway. **b** Comparative analysis of TNF/IFN-γ co-producing cell frequencies of conventionally gated $T_{EM}$ and $T_{EMRA}$ cells, and KLR/GPR56-based subsets upon short-term PMA/Ionomycin stimulation ($n = 7$, interleaved scatters with median bars). Functionality was evaluated according to the amount of cytokine production. **c** KLR/GPR56-based subset composition within classically gated CD4+ $T_{EM}$ and $T_{EMRA}$ cells ($n = 10$, mean ± SEM). Statistical analysis by one-way non-parametric Friedman's test with post-hoc Dunn's multiple comparison test. *$p < 0.05$, **$p < 0.01$, ***$p < 0.001$, ****$p < 0.0001$

cells as high, and $T_{EMRA}$ cells as exhausted cytokine producers (Fig. 4a, top row).

To this end we defined the following subsets within total CD4+ T cells: (1) KLRB1−KLRG1−GPR56−KLRF1−, (2) KLRB1+ KLRG1−GPR56−KLRF1−, (3) KLRB1+KLRG1+GPR56−KLRF1−, (4) KLRB1+KLRG1+GPR56+KLRF1−, and (5) KLRB1+ KLRG1+GPR56+KLRF1+. We analyzed each of their IFN-γ and

TNF production (Fig. 4a, mid-row) to compare it with that of classically gated T cells, $T_N$, $T_{CM}$, $T_{EM}$, and $T_{EMRA}$ cells. Indeed, the KLR/GPR56-based subset definition allowed categorization of cytokine production potential along CD4+ memory T-cell differentiation.

Interestingly, after primary acquisition of the initial marker KLRB1, the expression of this marker seemed to contribute little

to the cytokine expression potential as KLRB1+KLRG1+ GPR56−KLRF1− and KLRB1−KLRG1+GPR56−KLRF1−, as well as KLRB1+KLRG1+GPR56+KLRF1− and KLRB1−KLRG1+ GPR56+KLRF1− subsets displayed only minor differences in cytokine production potential. In fact, the KLRB1− subsets (Fig. 4a, bottom row) recapitulated the progressive acquisition of cytokine expression potential during memory T-cell differentiation and terminal exhaustion with acquisition of the KLRF1 marker (Fig. 4a). Based on these results, we conclude that the combinatory expression profile of KLRB1, KLRG1, GPR56, and KLRF1 allows a refined classification of memory T-cell subsets along their differentiation line and correlates to their functional state, judging from their cytokine expression potentials (low, medium, high, or exhausted, Fig. 4b). This refinement now facilitates the definition of the most potent cytokine-producing subsets; them being KLRB1+KLRG1+GPR56−KLRF1−, KLRB1− KLRG1+GPR56+KLRF1−, and especially KLRB1+KLRG1+ GPR56+KLRF1−, which contained the highest proportion of TNF/IFN-γ co-producing cells (Fig. 4b). Whereas no significant differences in proportion of TNF/IFN-γ co-producing cells between $T_{EM}$ and $T_{EMRA}$ cells could be detected, KLRB1+ KLRG1+GPR56+KLRF1− cells contained significantly more than KLRB1+KLRG1+GPR56+KLRF1+ cells. Furthermore, this progressive change in proportion of TNF/IFN-γ co-producing cells was not only observed upon PMA/Ionomycin stimulation, but also apparent when *Staphylococcus* enterotoxin B, tetanus toxoid (TT), or cytomegalovirus (CMV) peptides were used for re-stimulation (Supplementary Fig. 3). Interestingly, whereas TT-reactive T cells already accumulated in KLRB1+KLRG1−GPR56− KLRF1− cells, CMV-reactive TNF/IFN-γ co-producing cells were only detectable in KLRB1+KLRG1+GPR56−KLRF1− and KLRB1+KLRG1+GPR56+KLRF1− cells, most likely reflecting the frequency of antigen contact.

Having revealed that classically gated $T_{EM}$ cells contain less TNF and IFN-γ co-producing cells as compared with the most potent subsets of the KLR/GPR56 classification, we wondered whether $T_{EM}$ cells are in fact composed of different subsets according to our KLR/GPR56-based definition. Indeed, although the high cytokine-producing subsets made up the majority of $T_{EM}$ cells, populations with a low or exhausted functional state were also present (Fig. 4c), which may explain the overall lower cytokine production potential in $T_{EM}$ cells compared with KLRB1+KLRG1+GPR56+KLRF1− cells (Fig. 4b). Furthermore, classically gated $T_{EMRA}$ cells were composed of mainly exhausted populations (Fig. 4c) that therefore showed generally lower cytokine production potential (Fig. 4b). Thus, the refined classification of memory T cells according to the KLR/GPR56 scheme reveals functional heterogeneity in the classical $T_{EM}$ and $T_{EMRA}$ subsets.

**Inflammation shows increase in hepatic cytokine producers**. In recent years, it became clear that significant phenotypical and functional differences exist between circulating and intra-tissue T cells[27,28]. We therefore studied our newly defined memory T-cell surface marker panel on T cells derived from human liver tissue. First, we compared the proportions of CD4+ T cells displaying a classical $T_N$, $T_{CM}$, $T_{EM}$, and $T_{EMRA}$ phenotype between blood of healthy controls, and blood and liver from patients with inflammatory biliary and hepatic diseases. As expected, T cells from liver samples contained the lowest proportions of $T_N$ and $T_{CM}$ cells but highest of $T_{EM}$ cells (Fig. 5a). Interestingly, and somewhat unexpected, the proportion of $T_{EMRA}$ cells in some liver samples was lower than in the corresponding blood samples. Second, we performed single-cell gene expression profiling (genes listed in Supplementary Table 2) on sorted blood- and liver-derived $T_{CM}$, $T_{EM}$, and $T_{EMRA}$ cells of

patients with inflammatory liver diseases (Fig. 5b). Unsupervised hierarchical cluster analysis of selected candidate gene marker expression resulted in a separation of two main clusters, which differed in the proportion of *KLRB1*-, *GPR56*-, *NKG7*-, and *KLRF1*-expressing cells. The left cluster contained the majority of *KLRB1*-expressing cells and was dominated by $T_{CM}$ (blood and liver) cells with enrichment of liver $T_{EM}$ and $T_{EMRA}$ cells. In contrast, the majority of the blood $T_{EM}$ and $T_{EMRA}$ cells was contained within the right cluster, which showed a strong enrichment of *GPR56*-, *KLRF1*-, and partially *KLRG1*-expressing cells. This indicated that there was a qualitative difference between liver and blood-derived $T_{EM}$ and $T_{EMRA}$ cells. Indeed, the proportion of *GPR56*-, *KLRG1*-, and partially *KLRF1*-expressing $T_{EM}$ cells showed a tendency to be lower in liver samples, whereas the opposite was true for the proportion of *KLRB1*-expressing $T_{EM}$ cells (Supplementary Fig. 4).

These findings led us to investigate on protein level whether the proportions of T-cell subsets defined by the KLR/GPR56-based classification were different between blood and liver samples. Indeed, liver samples contained less of the low cytokine-producing T-cell subset KLRB1−KLRG1−GPR56−KLRF1− and had increased high cytokine-producing subsets KLRB1−KLRG1+ GPR56−KLRF1− and KLRB1+KLRG1+GPR56−KLRF1− (Fig. 5c). We also observed a reduction in the exhausted phenotypes KLRB1−KLRG1+GPR56+KLRF1+ and KLRB1+ KLRG1+GPR56+KLRF1+ in the liver compared with blood samples. These changes in subset composition accumulate to a generally increased pro-inflammatory functionality of CD4+ T cells in the liver of patients compared with that in the blood. In line with this, liver $T_{EM}$ and $T_{EMRA}$ cells showed a clear reduction of functionally exhausted KLRF1+ subsets as compared with their blood counterparts (Fig. 6a). Although the hepatic TNF and IFN-γ expression in the exhausted subsets was even lower than their blood-derived counterparts, all other subsets showed a generally increased cytokine production in the liver compared with the blood (Fig. 6b).

**Reduced TCRβ diversity in our proposed KLR/GPR56 pathway**. In order to provide further evidence for the molecular relationship and linear differentiation of our proposed populations, we analyzed their T cell receptor (TCR) clonotypes by sequencing of the TCRβ chains. Thus, we performed fluorescence-activated cell sorting (FACS)-based enrichment of all five populations (1 = KLRB1−KLRG1−GPR56−KLRF1−, 2 = KLRB1+KLRG1−GPR56− KLRF1−, 3 = KLRB1+KLRG1+GPR56−KLRF1−, 4 = KLRB1+ KLRG1+GPR56+KLRF1−, and 5 = KLRB1+KLRG1+GPR56+ KLRF1+) from peripheral blood of healthy individuals followed by a TCRβ chain-sequencing analysis.

Interestingly, the frequency of TCR clones of cells belonging to the KLRB1−KLRG1−GPR56−KLRF1− subpopulation (population 1, Fig. 7) was rather low, whereas a progressive increase in clonal frequency was observed for T cells belonging to the other populations (Fig. 7a). This indicates expansion of particular clonotypes, which is further supported by the decrease in TCRβ diversity along our proposed KLR/GPR differentiation path (Fig. 7b).

Furthermore, the proposed subpopulations do not differentiate completely independent from each other, as TCR profiles overlapped especially between the late populations 4 and 5, but also 3 (Fig. 7c), and TCR clones dominating in the late-stage differentiated populations 4 and 5 can be found in the early-stage populations 1 and 2 (Fig. 7d). Taken together, these results further provide evidence for a linear differentiation along and molecular relationships between the proposed populations.

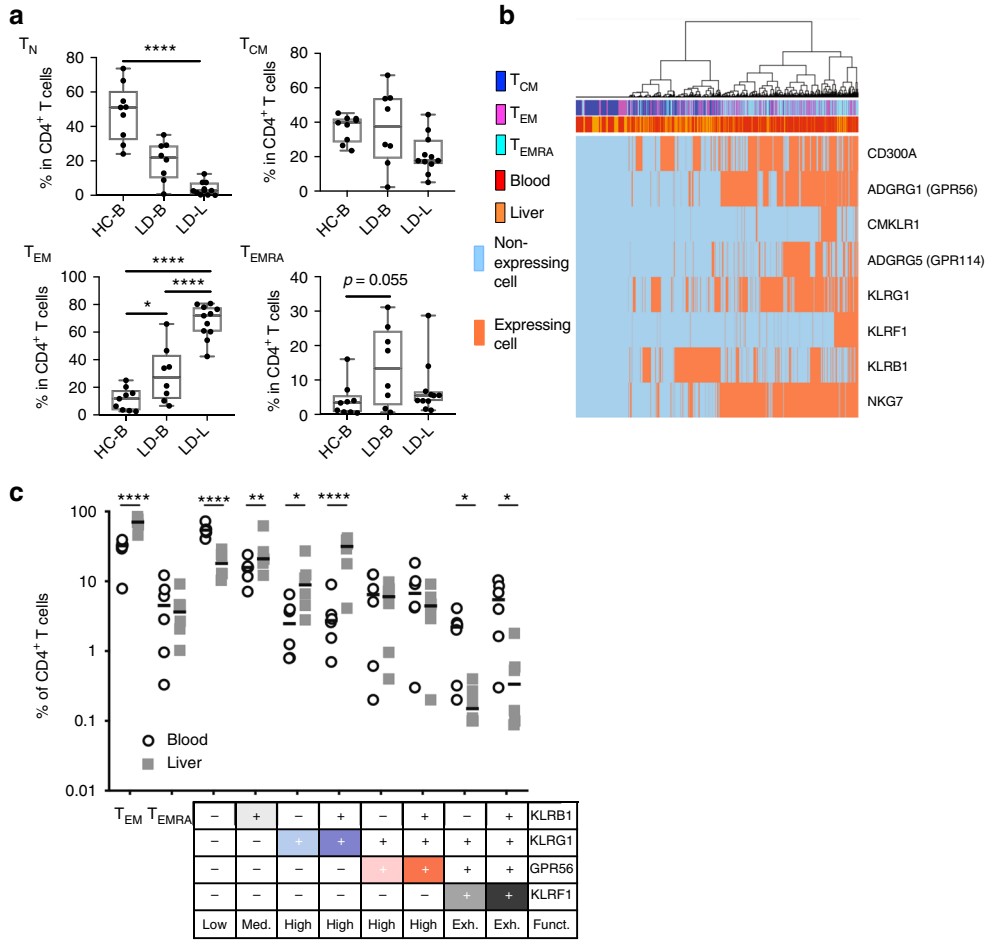

**Fig. 5** Decreased proportions of KLRF1-expressing CD4$^+$ memory T cells in the liver. **a** Frequency of $T_N$, $T_{CM}$, $T_{EM}$, and $T_{EMRA}$ cells within CD4$^+$ T cells of blood from healthy controls (HC-B, $n = 8$) as well as blood (LD-B, $n = 8$) and liver (LD-L, $n = 11$) of patients with inflammatory liver diseases. Whiskers extend to the minimum and maximum. $P$-values were determined by non-parametric Kruskal–Wallis test with post-hoc Dunn's multiple comparison test. $^*p < 0.05$, $^{****}p < 0.0001$. **b** Unsupervised hierarchical cluster analysis of candidate gene expression results at single-cell level in five blood and five liver samples of patients. In total, 152 (blood) and 86 (liver) CD4$^+$ $T_{CM}$, 249 (blood) and 82 (liver) $T_{EM}$, as well as 183 (blood) and 114 (liver) $T_{EMRA}$ cells were analyzed. Classification as expressing and non-expressing cells based on individual defined limit of detection (LoD) Ct values. Data are provided with the Source Data file. **c** Proportions of CD4$^+$ $T_{EM}$ and $T_{EMRA}$ cells and subsets according to KLRB1, KLRG1, GPR56, and KLRF1 protein expression within CD4$^+$ T cells from blood ($n = 6$) and liver ($n = 6$) of patients. Data are shown as individual scatter plots with median. Statistical analysis by one-way non-parametric Friedman's test with post-hoc Dunn's multiple comparison test. $^*p < 0.05$

**In vitro differentiation supports KLR/GPR56-based pathway.** Our results indicate that during human CD4$^+$ memory T-cell differentiation changes in TNF and IFN-γ production are a result of progressive acquisition of KLRB1, KLRG1, GPR56, and KLRF1 expression.

To further validate our proposed differentiation pathway, we performed in vitro experiments by sorting the first four subpopulations (1 = KLRB1$^-$KLRG1$^-$GPR56$^-$KLRF1$^-$, 2 = KLRB1$^+$KLRG1$^-$GPR56$^-$KLRF1$^-$, 3 = KLRB1$^+$KLRG1$^+$GPR56$^-$KLRF1$^-$, and 4 = KLRB1$^+$KLRG1$^+$GPR56$^+$KLRF1$^-$) from the peripheral blood of healthy individuals followed by polyclonal stimulation with anti-CD3/CD28 antibodies. Unfortunately, the cell number of the fifth population (5 = KLRB1$^+$KLRG1$^+$GPR56$^+$KLRF1$^+$) was too low to perform in vitro stimulation experiments.

The majority of the cells from the sorted populations kept their initial marker expression profile for 48 h, meaning that, e.g., only 2% of KLRB1$^+$KLRG1$^-$GPR56$^-$KLRF1$^-$ cells (starting population 2) became KLRB1- or that 20% of KLRB1$^+$KLRG1$^+$GPR56$^+$KLRF1$^-$ cells (starting population 4) lost GPR56 expression (Fig. 7e). In contrast to the other populations, subset 3 (KLRB1$^+$

KLRG1$^+$GPR56$^-$KLRF1$^-$) did show a higher degree of plasticity with ~40% of the cells becoming KLRB1$^+$KLRG1$^-$GPR56$^-$KLRF1$^-$. However, most importantly, the investigation revealed that the populations differentiate even further along the proposed path with, e.g., KLRB1$^+$KLRG1$^+$GPR56$^-$KLRF1$^-$ cells (starting population 3) acquiring GPR56 but also KLRF1 expression.

In addition, we analyzed and compared the intracellular TNF and IFN-γ expression of the in vitro differentiated subsets (Fig. 7f). Due to cell number limitations upon FACS-based enrichment, the experiments could be only done with the starting populations 1, 2, and 3. Upon in vitro differentiation, the subpopulations display the expected increase in proportions of TNF and IFN-γ double producers along the KLR/GPR56 pathway. The highest frequency was observed for KLRB1$^+$KLRG1$^+$GPR56$^-$KLRF1$^-$ and KLRB1$^+$KLRG1$^+$GPR56$^+$KLRF1$^-$ cells, whereas a decline was observed upon acquisition of KLRF1 expression.

Taken together, our findings introduce a surface marker classification scheme for CD4$^+$ memory T cells, which recapitulates their differentiation pathway and more precisely links quantity of TNF and IFN-γ production to each subset compared with the classical CCR7/CD45RA-based index. This might be of

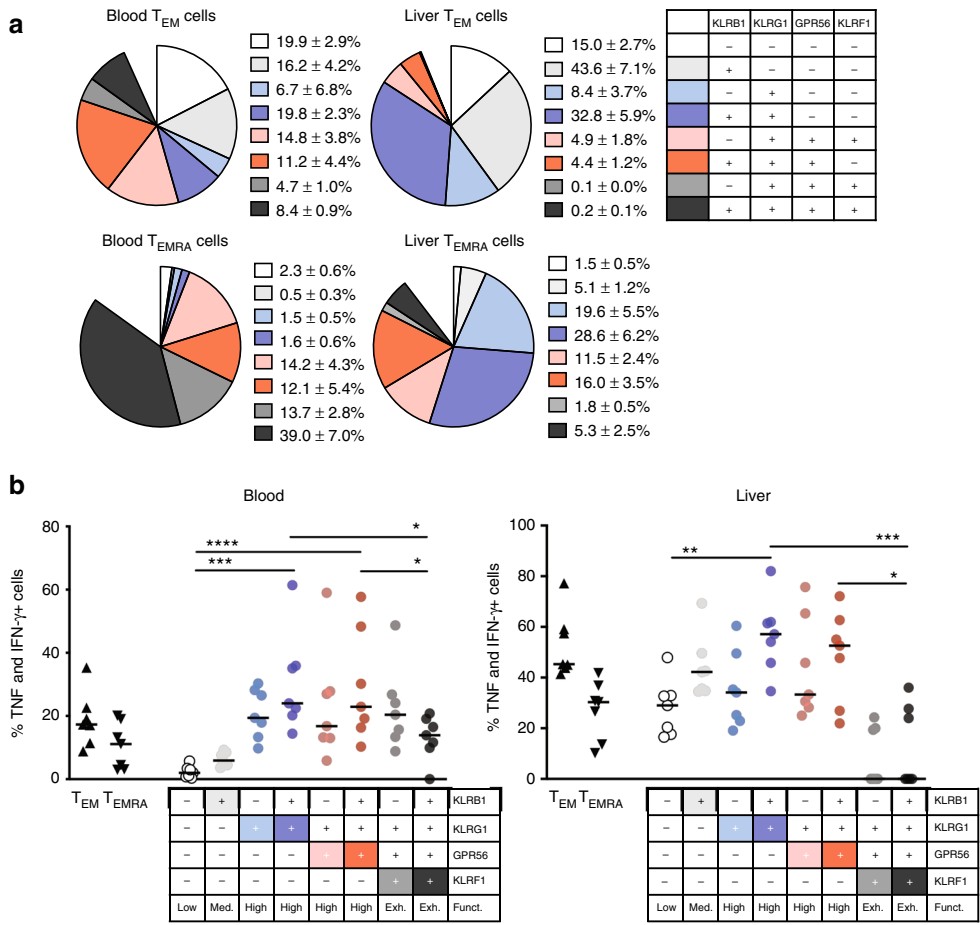

**Fig. 6** Increase in intrahepatic cytokine producers of TEM and TEMRA cells. **a** Subset composition gating within classically gated CD4$^+$ T$_{EM}$ and T$_{EMRA}$ cells of the blood and liver from patients with inflammatory liver diseases ($n = 7$, individual scatter plots with median). **b** Proportions of TNF/IFN-γ co-expressing cells upon pre-gating of T$_{EM}$, T$_{EMRA}$, or KLR/GPR56-based subsets following PMA/Ionomycin stimulation of PBMCs or liver leukocytes ($n = 7$ each). Statistical analysis was performed by one-way non-parametric Friedman's test with post-hoc Dunn's multiple comparison test. *$p < 0.05$, **$p < 0.01$, ***$p < 0.001$, ****$p < 0.0001$

particular interest for the characterization of T cells from diseased tissues, in which specific functional subsets might play an essential role for the pathophysiology and maintenance of disease.

## Discussion

Ever since the first categorization of CD4$^+$ or CD8$^+$ T cells into populations defining their differentiation status based on CD45RA and CCR7 expression, discussions arose around the overall validity[9]. Indeed, recent findings on functional classification of CD8$^+$ memory T cells have revealed that categorization based on CD62L or CCR7 expression, and thus lymph node homing properties, is not sufficient[29]. Therefore, it is not surprising that it was questioned whether the thereby defined subsets indeed represent homogeneous populations or whether individual cells differed greatly in their functional state[30–35], which for CD4$^+$ T cells is mainly defined by their cytokine expression potential.

We here addressed these questions for human CD4$^+$ memory T cells and assessed cellular heterogeneity on single-cell level within classically gated CD4$^+$ memory T lymphocytes from blood as well as from liver tissue of patients. As expected, we found pronounced heterogeneity within each subset on the overall transcriptional level, but also on the functional level assessed by single-cell cytokine secretion measurements. From these data, we developed a subset classification system based on the progressive acquisition of surface expression of the NK-cell-associated

proteins KLRB1, KLRG1, GPR56, and KLRF1. We show that this classification thoroughly mirrors the memory differentiation line and is superior in indicating the cytokine production potential of the individual subsets compared with the classical CD45RA/CCR7-based system.

Our findings on the progressive expression of multiple KLRs and GPR56 with final acquisition of KLRF1, resulting in a decline of cytokine production potential, is in line with published reports on murine CD8$^+$ and CD4$^+$ memory T cells. Analysis of phenotypic properties of murine CD8$^+$ and CD4$^+$ exhausted memory T cells revealed a correlation between decreased TNF/IFN-γ co-production potential and progressive expression of multiple inhibitory receptors, such as PD-1, LAG-3, 2B4 (CD244), and CD160 or PD-1, CTLA4, CD200, and BTLA, respectively.[36,37] However, investigations on human memory T cells and in particular CD4$^+$ memory T cells have been not performed so far.

Furthermore, the investigations on murine memory T cells were limited to the characterization of T cells with high effector function or non-functional exhausted T cells, and did not allow following the complete memory T-cell development. Incorporating single-cell gene expression profiling, Wanderlust analysis and TCRβ chain sequencing enabled us to propose an alternative path of human CD4$^+$ memory T-cell development defining populations with low, medium, high, and finally exhausted functional states. In our screen, expression pattern of KLRB1,

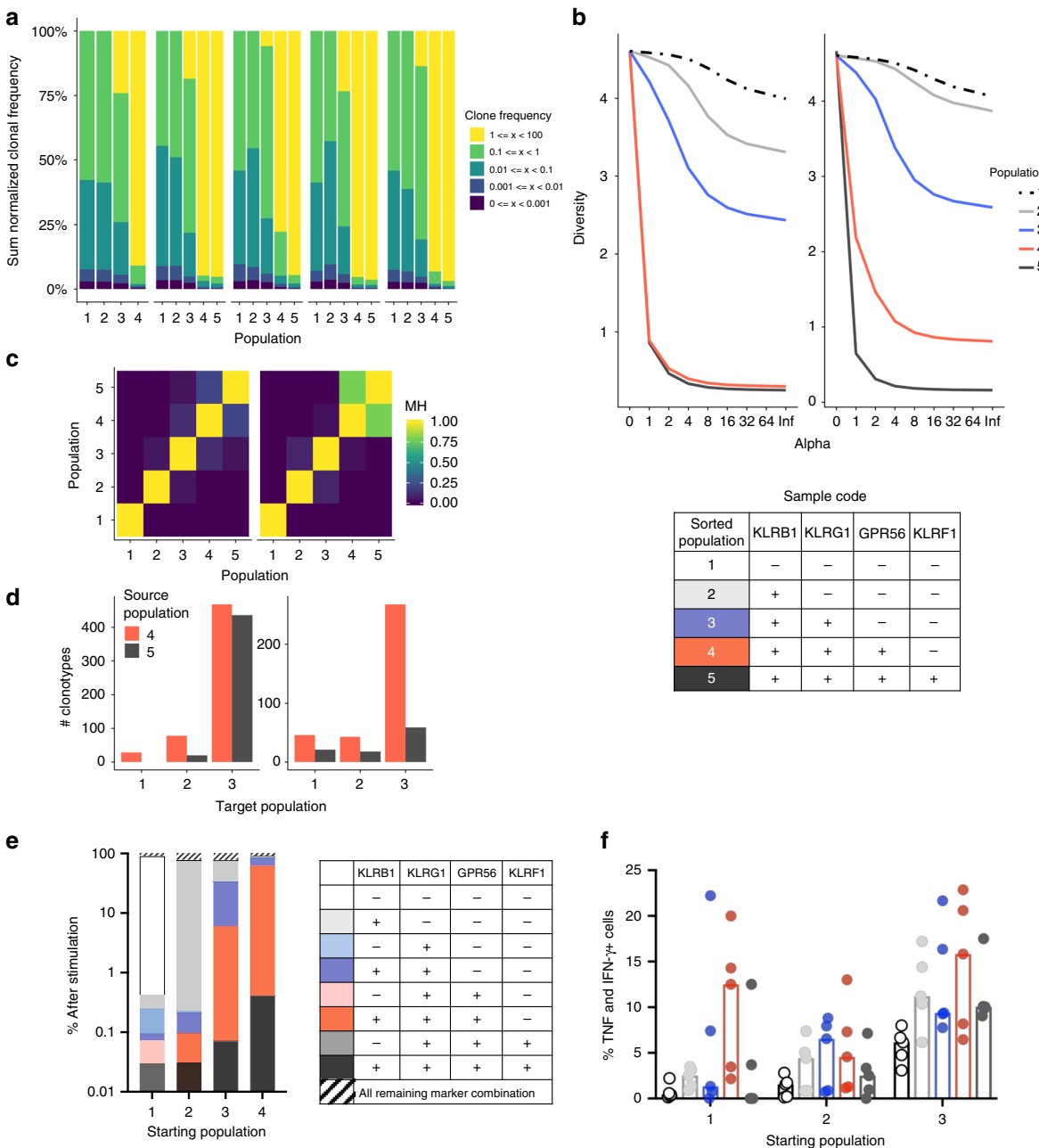

**Fig. 7** Clonotypic analysis and differentiation capacity of proposed CD4+ T-cell subsets. Clonal space (**a**), Rényi diversity profile (**b**, two exemplary), and clonal similarity (**c**, two exemplary) of TCRβ chain of sorted KLR/GPR56⁻ (population 1), KLRB1⁺ (population 2), KLRB1⁺KLRG1⁺ (population 3), KLRB1⁺KLRG1⁺GPR56⁺ (population 4), and KLRB1⁺KLRG1⁺GPR56⁺KLRF1⁺ (population 5) CD4⁺ T cells from healthy controls (n = 5). **d** Number of TCR clonotypes dominating in source population 4 and 5 identified in target populations 1, 2, and 3. The scaling factor Alpha of the Rényi diversity profile yields the sample diversity with different weighting of the clonotype proportion, see Methods. The clonotypes were verified prior to diversity calculation. The clonal similarity was assessed using the index of Morisita–Horn (1 indicates identity). Data are accessible within the European Nucleotide Archive (ENA) Accession Number PRJEB31283. **e** KLRB1, KLRG1, GPR56, and KLRF1 protein expression profile upon 48 h anti-CD3/CD28 mAb in vitro stimulation of indicated sorted CD4⁺ T-cell populations from PBMCs of healthy controls (n = 5). **f** Proportions of TNF/IFN-γ co-producing cells of in vitro differentiated populations upon 96 h anti-CD3/CD28 mAb in vitro stimulation of indicated sorted CD4⁺ T-cell populations from PBMCs of healthy controls (n = 5, interleaved scatters with median bars). Due to the low frequency of population 4 and 5 within PBMCs, cytokine analysis was only feasible for starting populations 1, 2, and 3

KLRG1, GPR56, and KLRF1 appeared to be most informative. Although PD-1 expression is described to be associated with T-cell exhaustion[38] and thus terminal differentiation of T cells, our initial RNA microarray analysis did not reveal a significant enrichment of PD-1 transcription within CD4⁺ T$_{EM}$ and T$_{EMRA}$ cells, as we also observed transcription in T$_{CM}$ cells. Furthermore,

our PD-1 protein expression analysis revealed a high expression in T$_{CM}$ and especially T$_{EM}$ cells but a reduction in T$_{EMRA}$ cells.

Our four identified surface markers, KLRB1, KLRG1, GPR56, and KLRF1, were all first described in relation to their high expression in NK cells[17,39–42], indicating similarities between NK cell differentiation and memory/effector T-cell development.

The C-type lectin KLRB1, also known as CD161, has been shown to be expressed by CD4$^+$ and CD8$^+$ T cells. For CD4$^+$ T cells, KLRB1 expression was mainly ascribed to IL-17-producing Th17 cells[25]. However, other recent publications identified also broader KLRB1 expression across different T-cell lineages expressing, e.g., IL-17 or TNF/IFN-γ, which is in agreement with our findings[43–45]. For NK cells, KLRB1 ligation was long thought to trigger only inhibitory signals[43]. However, a recent report revealed pro-inflammatory functions of KLRB1$^+$ NK cells[46]. Also, for T cells inhibitory and costimulatory roles have been proposed[43]. Indeed, our results also revealed a significant increase in cytokine-producing CD4$^+$ memory T cells with acquisition of KLRB1 expression. Interestingly, we did observe differences between KLRB1 transcription and protein expression. Whereas KLRB1 transcription was mainly limited to $T_{CM}$ and $T_{EM}$ cells, protein expression was observed for $T_{CM}$, $T_{EM}$, as well as $T_{EMRA}$ cells. In addition, $T_{EMRA}$ cells (Fig. 2a) positive for KLRB1 and the most exhausted KLRB1$^+$KLRG1$^+$GPR56$^+$KLRF1$^+$ cells showed reduced KLRB1 expression per cell.

The KLRG1 is a marker for T-cell senescence, as expressing cells have limited proliferative capacity[18,20]. However, KLRG1-expressing T cells are not exhausted, as they display cytokine production and cytotoxic potential[47]. KLRG1 expression is supposed to be limited to tissue-homing and thus $T_{EM}$ and $T_{EMRA}$ cells[8,28,48–50]. Our own data showed that also a significant proportion (≈22 %) of CCR7-expressing $T_{CM}$ cells were KLRG1$^+$. These findings are in agreement with other published reports showing that also $T_{CM}$ cells can express KLRG1, which was associated with increased production of effector cytokines[35]. Indeed, our results also revealed a dramatic increase of cytokine production potential as soon as the T cells acquired KLRG1 expression.

Already in the first report describing the NK cell triggering activity of KLRF1, also known as NKp80, its expression on a subset of T cells was observed[42]. In addition, it was shown that NKp80 ligation can augment CD3-stimulated degranulation and IFN-γ secretion by effector memory CD8$^+$ T cells[51]. This is contradictory to the here described results, as KLRF1 acquisition was associated with a decline in cytokine production potential. However, our investigations were performed on CD4$^+$ T cells and, at least in mice, different properties for CD4$^+$ in comparison with CD8$^+$ T cells were recently described[37]. It remains to be investigated whether KLRF1 plays an inhibitory role for human CD4$^+$ memory T-cell activation. Nevertheless, KLRF1 expression was able to identify CD4$^+$ memory T cells with reduced cytokine production potential regardless of cohort (healthy control vs. patient) or tissue-type origin.

GPR56 was shown to be expressed by cytotoxic NK and T lymphocytes including CD8$^+$, CD4$^+$, and γδ$^+$ T cells[41]. For NK cells, an inhibitory role for GPR56 in controlling steady-state activation by associating with the tetraspanin CD81 was revealed[52]. Similar to KLRF1, the role of GPR56 for stimulation-dependent production of cytokines by human CD4$^+$ T cells is unknown and needs to be investigated in further studies. In support of our findings, a recent report by Tian et al.[53] also showed that CD4$^+$ $T_{EMRA}$ cells are heterogeneous and can be subdivided into at least two subpopulations based on their GPR56 expression. In their publication, the authors focussed on the cytotoxic potential and did not study the cytokine production potential in greater detail. Furthermore, our results now show that GPR56 in conjunction with expression of KLRs is not limited to reveal functional heterogeneity of $T_{EMRA}$ cells in blood but also of $T_{EMRA}$ and $T_{EM}$ cells residing in the liver.

Our results on successful induction of, e.g., KLRG1 and GPR56 or even KLRF1 expression upon in vitro stimulation of sorted subsets further supports our hypothesis on progressive expression of killer-like receptors and GPR56, describing an initial increase and final decline in cytokine production potential. This is also in line with our previous findings on linear differentiation from $T_{CM}$, via $T_{EM}$, and towards $T_{EMRA}$ cells[54]. There we detected a progressive change in the epigenome of the cells, which could explain the progressive expression pattern described here.

Finally, our results with increased $T_{EM}$ but decreased or equal $T_{EMRA}$ frequencies in the liver in comparison with that in the blood is in agreement with recent descriptions on spatial mapping of human T-cell compartmentalization[11]. Although the authors did not investigate liver tissue, they did report increased $T_{EM}$ frequencies within intestinal and lung tissues compared with that in the blood, whereas $T_{EMRA}$ frequencies did not vary. However, the here reported combinational expression pattern of KLRs and GPR56 challenges the usefulness and gain of analyzing overall $T_{EM}$ and $T_{EMRA}$ frequencies, as analysis of KLR/GPR56 composition seems to be superior in discriminating between high and exhausted cytokine-producing cell subsets, with the exhausted subsets being decreased in intrahepatic CD4$^+$ $T_{EM}$ and $T_{EMRA}$ cells in comparison with their blood equivalents.

In summary, our data reveal that identifying human CD4$^+$ memory T-cell populations based on the expression pattern of KLRB1, KLRG1, GPR56, and KLRF1 enables a better definition of functional states especially in peripheral tissues as compared with the classical CD45RA/CCR7-based categorization. These findings improve understanding of CD4$^+$ memory T-cell development and function, and thus might have implications for clinical diagnostics and development of more target-specific immune therapies. It will be interesting to see whether the here described combinational expression profile and functional subsets might aid in improving prediction of disease progression in inflammatory diseases or in therapeutic efficacy upon vaccination or checkpoint inhibition.

## Methods

**Samples.** Heparinized blood and liver samples were collected from patients with written consent undergoing partial liver resection or hepatectomy followed by liver transplantation. Patients included suffered from following diseases: liver cirrhosis, hepatocellular carcinoma, liver metastases, and cancerous diseases of the biliary tract, including Caroli disease, gall bladder carcinoma, cholangiocellular carcinoma, and Klatskin tumor. Median age of the patients was 68 years, ranging from 52 to 81 years. Liver samples were taken from non-cancerous and non-necrotic parts of the resected liver tissue and were preserved in Hank's balanced salt solution (HBSS, Gibco, Thermo Fisher Scientific, UK). Heparinized blood from age-matched healthy individuals was collected. All samples were processed within an hour after retrieval.

Sample collection was performed following the Declaration of Helsinki, the European Guidelines on Good Clinical Practice, with permission from the relevant national and regional authority requirements and ethics committees (EA2/044/08, EA1/116/13, EA2/020/14, EA1/290/16, EA1/291/16 & EA1/292/16, Ethics Committee of the Charité Berlin). All patients gave written consent.

**Isolation of PBMC cells.** PBMC cells were isolated at room temperature by density gradient centrifugation (Biocoll, Biochrom, Germany) of heparinized blood diluted 1:2 in phosphate-buffered saline (PBS, Gibco, Thermo Fisher Scientific, UK). Cell number was determined using a hemocytometer. Isolated PBMCs were directly used for sorting, stimulation, or cryopreservation.

**Isolation of intrahepatic lymphocytes.** Liver tissue was dissected into 1 mm$^3$ fragments and digested with agitation (75–80 r.p.m.) at 37 °C for 30 min in a digestive solution containing 2% fetal calf serum (FCS, Biochrom), 0.6% bovine serum albumin (BSA), 0.05% collagenase type IV (Sigma-Aldrich, Germany), and 0.002% DNAse I (Sigma-Aldrich, Germany) per 1 g tissue and 10 ml. Undissociated tissue was pressed through a steel sieve and dissolved in same solution. Dissociated tissue in solution was centrifuged at 500 × g. Tissue components were diluted in HBSS. Tissue suspension was centrifuged at 30 × g to separate and discard the hepatocyte-rich matrix. Still undissociated tissue was removed by filtration through 100 µm nylon mesh, leaving a cell suspension. Hepatocytes were removed using a 33% Biocoll density gradient centrifugation. Cells were washed with HBSS

and red blood cell lysis was performed using distilled water. Isolated intrahepatic lymphocytes (IHLs) were cryopreserved in liquid nitrogen.

**Surface staining and T-cell subset sorting.** CD4+ T cells were enriched by magnetic activated cell sorting (MACS) using 20 μl of anti-CD4 MicroBeads (anti-CD4 microbeads, human, Miltenyi Biotec, Germany) and 80 μl MACS buffer (PBS with 0.5% BSA and 2 mM ethylenediaminetetraacetic acid) per $10^7$ cells. CD4+ T cells were stained for surface expression in MACS buffer at a concentration of $2 \times 10^8$ cells/ml. Cells were washed and stained with 4′,6-diamidino-2-phenylindole (1:250) and then sorted using the flow cytometer BD FACSAria II (BD Biosciences, Germany) into the following CD4+ T-cell subpopulations: Treg (CD25highCD127low) and non-Treg: TN (CD45RA+ CCR7+), TCM (CD45RA- CCR7+), TEM (CD45RA- CCR7-), and TEMRA (CD45RA+ CCR7-).

Antibodies used for surface staining were anti-CCR7 (GO43H7, 1:50), anti-CD25 (M-A251, 1:100), anti-CD45RA (HI100, 1:50), anti-CD127 (A019D5, 1:100), anti-CD3 (UCHT1, 1:200), anti-CD4 (OKT4, 1:200), anti-PD-1 (EH12.2H7, 1:20), anti-TIGIT (A15153G, 1:20), anti-TIM-3 (F38–2E2, 1:20), and LAG-3 (11C3C65, 1:20) from BioLegend, and anti-KLRB1 (191B8, 1:10), anti-KLRF1 (4A4.D10, 1:50), and anti-KLRG1 (REA261, 1:10, 1:50 since 2018/06 due to more concentrated formulation) from Miltenyi Biotec. See also Supplementary Fig. 5 for the pre-gating strategy.

**RNA microarray analysis.** Total RNA from sorted T-cell populations was isolated using TRIzol (Thermo Fisher Scientific, Bremen, Germany). RNA quality and integrity were determined using the Agilent RNA 6000 Nano Kit on the Agilent 2100 Bioanalyzer (Agilent Technologies). RNA was quantified by measuring light absorbance at 260 nm on a spectrophotometer (NanoDrop Technologies).

Sample labeling was performed as detailed in the One-Color Microarray-Based Gene Expression Analysis protocol (version 6.6, part number G4140–90040). Briefly, 10 ng of each total RNA samples was used for the amplification and labeling step using the Agilent Low Input Quick Amp Labelling Kit (Agilent Technologies). Yields of cRNA and the dye-incorporation rate were measured with the ND-1000 Spectrophotometer (NanoDrop Technologies).

The hybridization procedure was performed according to the One-Color Microarray-Based Gene Expression Analysis protocol (version 6.6, part number G4140–90040) using the Agilent Gene Expression Hybridization Kit (Agilent Technologies). Briefly, 1.65 μg Cy3-labeled fragmented cRNA in hybridization buffer was hybridized overnight (17 h, 65 °C) to Agilent Whole Human Genome Custom Oligo Microarrays 4 × 44 K (AMADID 014850) using Agilent's recommended hybridization chamber and oven. Following hybridization, the microarrays were washed once with the Agilent Gene Expression Wash Buffer 1 for 1 min at room temperature followed by a second wash step with preheated Agilent Gene Expression Wash Buffer 2 (37 °C) for 1 min. The last washing step was performed with acetonitrile.

**Single-cell gene expression analysis.** The C1 Single-Cell Auto Prep System (Fluidigm, South San Francisco, CA, USA) was used for single-cell isolation and pre-amplification to prepare separate single-cell cDNA in a 5–10 μm C1 Single-Cell PreAmp Integrated Fluidic Circuit (IFC) within a C1-Chip. For single-cell isolation, a cell suspension of at least 660,000 cells/ml was used, which enabled at least 2000 cells to enter the C1-chip. Visualization of cell loading (empty, single, doublets, or debris) was done using a light microscope. Single-cell capture rates were documented. Cell lysis, reverse transcription, and pre-amplification were performed on the C1-chip. cDNA of each cell was collected for qRT-PCR preparation. cDNAs and 48 TaqMan gene expression assays (Thermo Fisher Scientific), including *B2M* (beta-2-microglobulin) and an RNA spike-in (spike 1) as control values, were applied to the BioMark Gene Expression 48.48 IFC for gene expression analysis. Information on all TaqMan gene expression assays used are listed in Supplementary Table 2. Detailed workflow can be found in the Fluidigm Real-Time PCR User Guide (PN 68000088).

**Cell stimulation and intracellular cytokine staining.** Freshly isolated or thawed PBMCs or IHLs ($5 \times 10^6$) were stimulated with phorbol myristate acetate (50 ng/ml, Sigma-Aldrich, Germany) and ionomycin (1 μg/ml, Biotrend, Germany) for 6 h, Tetanus Toxid (40 LF/ml, AJ Vaccines, Denmark), or Staphylococcal enterotoxin B (100 μg/ml, Sigma-Aldrich, Germany) and anti-CD28 (1 μg/ml, BD Pharmingen, Germany) for 24 h (37 °C, 5 % CO₂).

CD4+ T cells enriched from freshly isolated PBMCs from healthy donors (CD4 MicroBeads, Miltenyi Biotec) were sorted for indicated marker expression using a BD FACSAria II. Sorted cells were stimulated with plate-bound anti-CD3 (1 μg/ml, BD Pharmingen, Germany) and soluble anti-CD28 (2 μg/ml, BD Pharmingen, Germany) in 96-well plates with $1 \times 10^5$ cells/well and treated with TGF beta RI kinase inhibitor II (50 ng/ml, Calbiochem, Merck, Germany) for 72 h. In all stimulation cultures, Brefeldin A (10 μg/ml, Sigma-Aldrich, Germany) was added during the last 4 h.

Cells were washed once with PBS and stained with Zombie UV Fixable Viability Kit (BioLegend, USA) for 15 min. Cell surface staining was performed as described above. Afterwards, cells were fixed and permeabilized (BD Cytofix/Cytoperm Fixation and Permeabilization Solution, BD Biosciences, Germany) for 20 min.

After washing twice with Perm/Wash buffer (BioLegend, USA), cells were stained intracellularly for 30 min. Antibodies used for intracellular staining were anti-TNF (Mab11, 1:200), anti-IFN-γ (4 S.B3, 1:100), anti-IL-4 (MP4–25D2, 1:50), anti-IL-17A (BL168, 1:20), anti-CD3 (OKT3, 1:100) (all BioLegend), and GPR56 (REA467, Miltenyi Biotec, 1:10). Samples were washed and acquired on a BD LSRFortessa (BD Biosciences, Germany). Data analysis was performed using FlowJo software version 10.1 (FlowJo, LLC, Ashland, OR, USA).

**Bioinformatical analysis of flow cytometry data.** To generate and visualize wanderlust trajectories of developmental changes in marker expression of CD4+ T cells, we used the Matlab Cyt toolbox[26]. The algorithm was run on CD8- non-Treg (exclusion of CD4+CD25highCD127low cells) pre-gated PMA/Ionomycin stimulated samples. In order to apply Wanderlust to samples where all gradual differentiation states are present, FCS files were selected to have a high proportion of TEMRA cells (5 healthy individuals) and density-dependent down-sampled to a total 50,000 cells using the R SPADE package[55].

Utilizing the Cytobank viSNE tool, t-SNE maps were generated for CD8- pre-gated T cells (w/o CD25highCD127low cells) of PMA/Ionomycin-stimulated samples, allowing for visualization of the phenotypic and functional heterogeneity at single-cell level[56,57]. CCR7, CD45RA, KLRB1, KLRF1, KLRG1, GPR56, TNF, and IFN-γ were selected for both Wanderlust and t-SNE dimension reduction.

**Bioinformatical analysis of RNA microarray data.** Fluorescence signals of the hybridized Agilent Microarrays were detected using Agilent's Microarray Scanner System G2505C (Agilent Technologies). The Agilent Feature Extraction Software (FES) 10.7.3.1 was used to read out and process the microarray image files.

After quantile normalization, pair-wise *t*-tests were conducted between TEMRA and TEM, TCM, and TN, and between TEM and TCM, and TN, respectively. The intersection analysis of TEMRA vs. TN, TEMRA vs. TCM, and TEMRA vs. TEM ( = TEMRA vs. TN/TCM/TEM) included genes with an at least twofold upregulation ($p < 0.05$) in the TEMRA group in each of the pair-wise comparisons. The selection TEMRA/TEM vs. TN/TCM included genes with an at least twofold upregulation ($p < 0.05$) in each of the comparisons TEMRA vs. TN, TEMRA vs. TCM, TEM vs. TN, and TEM vs. TCM. Genes expressed on the cell surface were extracted from the human cell surfaceome described in da Cunha et al.[58].

**Bioinformatical analysis of single-cell qRT-PCR results.** qPCR data were extracted by using the Fluidigm Real-Time PCR Analysis Software. R, version 3.1, was used for the statistical and hierarchical cluster analysis (R_Core_Team. R: A language and environment for statistical computing. R Foundation for Statistical Computing, Vienna, Austria URL http://www.r-projectorg/ 2014). The data contained the gene expression level (Log₂Ex), calculated by subtracting gene Ct values from limit of detection (LoD) Ct. Default LoD is preset to Ct 24. By analyzing the distribution of the gene expression level of each nest for each gene, we adjusted the LoD of certain genes to only include noise-free data. Samples with undetectable B2m and Spike 1 expression, as well as genes expressed by <10 cells were excluded. Heatmaps for single-cell Log₂Ex data were generated using Euclidean distance measure and Wards method for agglomerative hierarchical clustering. Binary heatmap coloring was chosen to indicate positive and negative marker expression according to the individual LoDs.

**TCRβ chain-sequencing and data analysis.** Next-generation sequencing was performed for TCR repertoire analysis. Recombined TCR-b locus was amplified as previously described[59]. Sequencing library preparation with consequent sequencing was performed using Illumina MiSeq Technology at the Genomics Unit at Centre for Genomic Regulation (Barcelona, Spain). Reads with an average quality score below 30 were excluded from the analysis. The remaining high-quality reads were processed using IMSEQ[60]. Each clonotype was assigned an ID including Vβ- and Jβ-gene identity, as well as CDR3 amino acid sequence.

The similarity of the top 100 expanded clones of two populations was determined by the Morisita–Horn similarity index[61], which considers the abundance of each sequence in each population. The index is given by:

$$S_{\mathrm{MH}} = \frac{2 \sum_{n=i}^{n_{PQ}} p_i q_i}{\left( \frac{\sum_{n=i}^{n_{PQ}} p_i}{P^2} + \frac{\sum_{n=i}^{n_{PQ}} q_i}{Q^2} \right) PQ} \tag{1}$$

where $P$ is the total number of sequences in one population, $Q$ is the total number of sequences in the second population, and $n_{PQ}$ is the total number of unique sequences in the two populations; $p_i$ is the proportion of the $i$th sequence in population $P$; $q_i$ is the proportion of the $i$th sequence in population $Q$. The index ranges from 0 to 1, where 0 is total dissimilarity and 1 is identical populations.

Clonal diversities of the TCRβ repertoires were evaluated for the top 100 expanded clones using Rényi diversity profiles[62]:

$$H_{\alpha} = \ln \left( \sum_{n=1}^{n} p_i^{\alpha} \right) \frac{1}{1 - \alpha} \tag{2}$$

$H_{\alpha}$ being the entropy; $n$ the total number of unique sequences; $p_i$ the proportion of the $i$th sequence, and $\alpha$ a scaling parameter. By varying the $\alpha$ parameter, different

diversity indices are calculated, such as the logarithm to the reciprocal Simpson diversity index at $\alpha = 2$. At $\alpha = 1$, the Rényi diversity is approximated by the Shannon diversity index. At $\alpha = 0$ and $\alpha = \infty$, the profiles provide the logarithm of richness and the logarithm of the reciprocal to the proportion of the most abundant sequence (Berger–Parker Index), respectively. This means that the sample with the highest value at $\alpha = 0$ has the highest richness, but that the lower value at $\alpha = \infty$ indicates higher proportion of the most abundant sequence. A sample with a profile that is overall higher than the profiles of other samples is more diverse. Conversely, if the profiles cross at one point no ranking in diversity can be performed.

**Statistics**. To test for differences in frequencies of T-cell subsets transcribing (single-cell qRT-PCR) the gene marker or expressing them at protein level (flow cytometry), a two-way analysis of variance and Sidak's multiple comparison test were performed. Differences in proportions of cytokine producers (paired samples) were tested using the non-parametric matched-pairs Friedman's test and post-hoc Dunn's multiple comparison test. $P < 0.05$ was considered statistically significant. Statistical analyses were calculated with GraphPad Prism 6.00 or R v3.3.1.

## Data availability

The TCR sequencing FASTQ data have been deposited in the European Nucleotide Archive (ENA) with the accession code PRJEB31283. The RNA microarray data have been deposited in the National Center for Biotechnology Information Gene Expression Omnibus (GEO)[63,64] and are accessible under the GEO series accession number GSE102005 . All data generated by single-cell rtPCR or flow cytometry are available in the Source Data file that contains the raw data for Fig. 1c & d, 2b, 3a, 4b & c, 5a, b & c, 6a & b, 7e & f, Supplementary Fig. 2a, 3, and 4.

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

## Acknowledgements

We thank the BCRT Flow Cytometry Lab (BCRT-FCL) for assistance with cell sorting. This work was supported by the Deutsche Forschungsgemeinschaft (SFB650) and Bundesministerium für Bildung und Forschung (e:Kid).

## Author contributions

K.-L.T. and S.S. performed the experiments, analyzed the data, and wrote the manuscript. D.B., K.S. and N.S. helped with the experiments. K.V. assisted with C1 experiments. C.A., C.I. and M.S. performed or helped with the in vitro stimulation and flow cytometry experiments. U.S. and N.B. performed and analyzed the next-generation sequencing-based TCRβ sequences. G.G., C.M. and J.K.P. discussed, interpreted the data, and edited the manuscript. S.T. performed the bioinformatics and statistical analysis of the microarray data. A.P., I.S. and U.G. helped in collecting patient samples and clinical data information. K.S. and C.I. helped editing the manuscript. B.S. designed and supervised experiments, discussed and interpreted the data, and contributed to the writing and editing of the manuscript.

## Additional information

**Competing interests:** The authors declare no competing interests.

