## [Peer Review File · Nature Communications]

Reviewers' comments:

Reviewer #1 (RNA-seq, systems biology)(Remarks to the Author):

In this manuscript, entitled "Successive expression of killer-like receptors and GPR56 defines the cytokine production potential of human CD4+ memory T cells", written by Truong et al., the authors characterized the cellular heterogeneity of the T cells. For this purpose, they employed a series of single cell techniques such as FACS and single cell qPCR. They clarified that canonical categorization of T cells, which separates naïve, central memory, and effector memory subpopulations based on the CCR7 and CD45RA expression patterns, is not sufficient to explain the production ability of the key cytokines, IFN-gamma, TNF-alpha and so on. Instead, based on the results of this paper, the authors propose a novel classification criterion, that is, by examining the expressions of KLRB1, KLRG1, GPR56, and KLRF1, the T cells can be more clearly separated into the subclasses. By doing so, they say the cytokine production ability could be better explained. The authors further examined the detailed expression patterns using a new analytical method, "Wanderlust". They found that KLRB1 starts to be expressed at the earliest differentiation stage, and then expressions of GPR56 and KLRG1 follow. Eventually KLRF1 is expressed and the T cells expressing LFRF1 have the lowest cytokine production ability. Generally, this is a well-written paper, conveying very interesting results. Indeed, the previously used canonical categorization has been found to be insufficient to correlate the T cell population with their cytokine production ability and other functional features. This imposes substantial problem when we try to make use of the immune system for various clinical purposes, such as immune-check point inhibition in cancers. Toward such a goal, I expect the strategy proposed here should be extremely useful. Followings are some points which I would like to suggest to further strengthen the contents of the paper.

Miscellaneous points:

1) The single cell gene expression analysis conducted here could be scaled to the transcriptome-wide analysis. Particularly, taking advantage of the C1 system, which the authors used in this study, further extensive sequencing analysis of the representative individual cells seems not so difficult using the pre-existing library pools. Such an analysis would give far richer information for the T cell heterogeneity and the molecular mechanisms. The single cell transcriptome analysis using Chromium system would be further complementary to enrich the data contents.

2) Please explain in more details how the t-SNE plot was drawn. How the cells were clustered is important to interpret the heat map results. Particularly, I wonder how the plot should be correlated with the single cell gene expression results, and how the results were integrated to the following "Wanderlust" analysis. Those points could be described in a more precise but non-expert-friendly manner. CyTOF analysis could be also considered to be conducted to validate the results.

3) In Figure 7, the authors showed that KLRB1 single positive cells give arise to KLRB1+KLRG1+ and KLRB1+KLRG1+GPR56+. Here, they only showed the bar plots, and it looks like, if we sum up the figures, it does not reach 100. I wonder where the other fractions, if any, have gone. It would be beneficial to show usual dot plots of the FACS data to clarify what kind of populations are there, for example, to represent whether the KLRB1 negative cells may exist or not.

4) Related to the previous point, the authors examined the expression patterns of the limited combination (namely KLRB1 vs KLRG1 and KLRF1 vs GPR56) of the four markers in Figure 2. It would be informative to show other patterns to see if any other populations than the proposed path (in Figure 4) exist. The transcriptome-wide analysis as described above would be even better.

5) Further related to the previous points, if the remaining approximately 70 % of the cells, as

presented in Figure 7, had already lost the KLRB1 expression, this would lead to a serious question on the robustness of the proposed classification scheme. To what extent are the proposed populations, such as KLRB1+, KLRB1+KLRG1+ and so on, likely to have certain plasticity? In other words, the robustness of the proposed path as shown in Figure 4 should be further evaluated. For example, if the cells in a population such as KLRB1+KLRG1+GPR56+KLRF1- are sorted and stimulated, can we expect if all of the cells would turn out to be “quadruple-positive” population. Alternatively, would some cells be branched off from the path?

6) The authors discussed the PD-1 expression in the canonical categorization (in Discussion section). What about the expression of the other markers for exhaustion such as TIM-3, LAG-3 and so on in the proposed populations?

7) Indeed, I could not clearly see how the results should be compared between this study in humans and previous studies mainly in mice. Providing a concise table in supplementary information with this regard would be extremely helpful.

8) I sincerely hope the authors further pursue the path to elucidate the molecular mechanisms of the gene expression changes, also in conjunction with other previous knowledge. I believe such study should be no less important than to investigate the clinical relevance of the new findings described here. After all, we would not be able to totally rely on the phenomenon for which the mode-of-action still remains elusive

Reviewer #2 (Memory T cell biology)(Remarks to the Author):

In this manuscript Truong et. al. put forth an alternative classification scheme for CD4+ memory T cells based on surface markers related to cytokine production instead of the classical TCM, TEM, TEMRA divisions. They use cytokine production to define surface markers within the TEM and TEMRA populations relevant to ‘high producers’, or conversely, ‘exhausted’ cells. They observe KLRF1 to be the most relevant marker to identify exhausted CD4 T cells whereas KLRB1 is associated with high cytokine production. Additionally, they assign a differentiation scheme in which acquisition of NK-associated markers KLRB1, KLRG1, GPR56, and KLRF1 occurs in a successive and linear fashion leading to first improved, then declining function. There are several significant issues with the manuscript which need to be addressed before it should be considered for publication. Specific points are outlined below.

Specific points:

1. What is the rationale for combining TEM and TEMRA in Figure 1B when most of the paper keeps these two as separate cell groups? What is the scale for the heat map in Figure 1B?
2. (Figure 1C, D); 200-400 cells were analyzed by qRT-PCR, followed by a cut off of 10 cells expressing a gene to then perform clustering analysis. 4 donors were used in total. This is a very low number of samples and cells to score as a significant finding, particularly given that strong heterogeneity in the TEMRA pool between donors has been noted before.
3. An issue with this manuscript is that it's sometimes hard to follow. There are so many permutations of the data displayed that the main point(s) get lost. I would suggest limiting the data display (or moving more to supplemental) to make the main points more evident. For example, in Figure 3, remove IL-4 and IL-17A as cells display minimal to no production. Figure 5 is also hard to digest in its current form.
4. The evidence to indicate the linear progression of cells from KLRB1+ to KLRB1+KLRG1+ and so on is not convincing. There is a lot of emphasis on Figure 3C for this claim (Wanderlust analysis), but that

makes assumptions on linear differentiation from the onset. The authors then assume this analysis is correct and draw arrows in Figure 4 to indicate how the differentiation pathway relates to cytokine production.

5. Figure 7 provides some evidence to support their differentiation theory, but it falls short of being convincing. Where are the other 80% of cells in this assay? What happens if you start with KLRB1+KLRG1+ cells, KLRF1-expressing cells, etc.? This is a critical experiment to support the underlying differentiation conclusion which is central to the paper. In 7B, there is no negative control to compare the cytokine production to, nor are single positive KLRB1-expressing cells included.

6. Throughout the manuscript the authors indicate Cxc3cr1 cells. Is this supposed to be CX3CR1?

First of all, thank you very much for evaluating our manuscript. We would also like to express our thankfulness to the reviewers for their helpful and encouraging comments. We addressed all comments raised by the reviewers and revised the manuscript accordingly.

In particular, we provided further evidence supporting the linear differentiation of human CD4⁺ memory T cells according to our marker combinations.

- 1) We have performed TCR-sequencing of our FACS-enriched T cell populations and can show that the TCR diversity declines along the proposed differentiation pathway, but that the TCR clonotypes overlap and clonotypes dominating in the late stages can be found in earlier differentiation stages, though at much lower frequencies. These data provide strong molecular evidence for the relationship of our proposed populations and that differentiation of individual CD4⁺ T cell clones is associated with a successive expression of killer-like receptors and GPR56.
- 2) We have also extended our *in vitro* stimulation experiments of FACS-enriched memory CD4⁺ T cell subpopulations and can demonstrate that the sorted subsets do not substantially lose marker expression but rather that they acquire expression of additional marker and further differentiate along our proposed pathway from KLR / GPR56 negative, KLRB1⁺ single positive, KLRB1⁺KLRG1⁺ double positive, KLRB1⁺KLRG1⁺GPR56⁺ triple positive to KLRB1⁺KLRG1⁺GPR56⁺KLRF1⁺ quadruple positive T cells. This is associated with changes in cytokine expression potential similar to that of *ex vivo* analyzed populations.

Furthermore, we have increased the sample numbers of the single cell gene expression analysis and analyzed additional T_{EM} (n=61) and T_{EMRA} (n=50) cells. Unsupervised cluster analysis of these additional cells resulted in nearly identical expression patterns validating our previous results justifying our conclusions.

Please find enclosed the revised version of the manuscript and below a separate itemized series of responses to the comments.

We believe that our manuscript has significantly improved following the helpful comments and thank you for kindly considering the revised version of our manuscript. Please do not hesitate to contact us if we can be of any further help.

Reviewer #1 (RNA-seq, systems biology) (Remarks to the Author):

In this manuscript, entitled “Successive expression of killer-like receptors and GPR56 defines the cytokine production potential of human CD4+ memory T cells”, written by Truong et al., the authors characterized the cellular heterogeneity of the T cells. For this purpose, they employed a series of single cell techniques such as FACS and single cell qPCR. They clarified that canonical categorization of T cells, which separates naïve, central memory, and effector memory subpopulations based on the CCR7 and CD45RA expression patterns, is not sufficient to explain the production ability of the key cytokines, IFN-gamma, TNF-alpha and so on. Instead, based on the results of this paper, the authors propose a novel classification criterion, that is, by examining the expressions of KLRB1, KLRG1, GPR56, and KLRF1, the T cells can be more clearly separated into the subclasses. By doing so, they say the cytokine production ability could be better explained. The authors further examined the detailed expression patterns using a new analytical method, “Wanderlust”. They found that KLRB1 starts to be expressed at the earliest differentiation stage, and then expressions of GPR56 and KLRG1 follow. Eventually KLRF1 is expressed and the T cells expressing LFRF1 have the lowest cytokine production ability. Generally, this is a well-written paper, conveying very interesting results. Indeed, the previously used canonical categorization has been found to be insufficient to correlate the T cell population with their cytokine production ability and other functional features. This imposes substantial problem when we try to make use of the immune system for various clinical purposes, such as immune-check point inhibition in cancers. Toward such a goal, I expect the strategy proposed here should be extremely useful. Followings are some points which I would like to suggest to further strengthen the contents of the paper.

Miscellaneous points:

1) The single cell gene expression analysis conducted here could be scaled to the transcriptome-wide analysis. Particularly, taking advantage of the C1 system, which the authors used in this study, further extensive sequencing analysis of the representative individual cells seems not so difficult using the pre-existing library pools. Such an analysis would give far richer information for the T cell heterogeneity and the molecular mechanisms. The single cell transcriptome analysis using Chromium system would be further complementary to enrich the data contents.

We like to thank the reviewer for this comment. We did indeed start our investigations with a whole transcriptome-based analysis of sorted CD4+CD45RA+CCR7+ T_N, CD4+CD45RA-CCR7+ T_{CM}, CD4+CD45RA-CCR7- T_{EM} and CD4+CD45RA-CCR7+ T_{EMRA} cells from human peripheral blood (see figure 1A & 1B). This led to the identification of candidate surface marker, whose gene expression is increased in CD4⁺ T_{EM} and especially T_{EMRA} cells in comparison to T_N and T_{CM} cells. The whole transcriptome results are accessible under the GEO series accession number GSE102005 (<http://www.ncbi.nlm.nih.gov/geo/query/acc.cgi?acc=GSE102005>).

For those candidate surface markers we performed single cell gene qRT-PCR on the C1 system to reveal whether some of them are heterogeneously expressed and define subsets within CD4⁺ T_{EM} and T_{EMRA} cells. Indeed, those surface markers identified T cell subsets, which differ in cytokine production potential at protein level.

We totally agree with the reviewer that additional transcriptome-based analysis of individual T cell subsets using our newly identified differentiation markers will provide further details into T cell

heterogeneity and differentiation pathways. However, for our single cell analysis we performed the targeted qRT-PCR-based technique which involves gene-specific cDNA preamplification but not cDNA library generation. Thus, with the available samples we cannot perform single cell transcriptome analysis.

Such studies are planned. However, in addition to space restrictions the representation of further transcriptome-wide analysis of representative T cell subsets would go beyond the scope of this manuscript. Therefore, we would kindly suggest, to present those data in a separate manuscript.

2) Please explain in more details how the t-SNE plot was drawn. How the cells were clustered is important to interpret the heat map results. Particularly, I wonder how the plot should be correlated with the single cell gene expression results, and how the results were integrated to the following “Wanderlust” analysis. Those points could be described in a more precise but non-expert-friendly manner. CyTOF analysis could be also considered to be conducted to validate the results.

We apologise if the explanation of t-SNE and Wanderlust analyses were not detailed enough.

In order to clarify the generation and interpretation of the t-SNE plots, and also how they correlate with the single cell gene expression results, we herewith provide a more detailed description. Sentences marked in yellow have been either modified or newly added:

The t-SNE plots were drawn to visualize co-expression patterns of the surface markers with TNF- α and IFN- γ expression. PBMCs of healthy controls were stimulated with PMA/Ionomycin and stained with fluorochrome-conjugated antibodies recognising CD3, CD4, CD25, CD127, CCR7, CD45RA, KLRB1, KLRG1, GPR56, KLRF1, TNF- α , IFN- γ , IL-4 and IL-17A.

T-SNE maps were created by arranging all conventional CD4⁺ T cells (pre-gating of CD3+CD4+ cells with exclusion of CD25^{high}CD127^{low} regulatory T cells) according to their similarity in surface marker (CCR7, CD45RA, KLRB1, KLRG1, GPR56, KLRF1) and cytokine expression (TNF- α , IFN- γ , IL-4, IL-17A).

The two-dimensional shape of all t-SNE plots is based on the overall similarities between the acquired conventional CD4⁺ T cells.

The colour code of each t-SNE plot reflects the distribution of marker positive and negative T cells, which e.g. allows the identification of CCR7 negative CD45RA negative or positive T_{EM} and T_{EMRA} cells in the upper left t-SNE plots.

We highlighted the area of CD4⁺ T_{EM} and T_{EMRA} cells within the plots (encircled black area) judging from their CD45RA and CCR7 expression pattern.

T-SNE plots showing the distribution of TNF- α and IFN- γ positive and negative cells revealed that expression of both cytokines is common but clearly heterogeneous within the T_{EM}/T_{EMRA} area, with certain subtypes being completely devoid of cytokine expression potential (blue arrows). In accordance with the single cell gene expression results we detected a homogenous KLRG1 expression in nearly all cells within the T_{EM}/T_{EMRA} area, whereas expression of the other marker is very heterogeneous.

Surprisingly, most cells in this cytokine-low area express all four surface markers with KLRF1 displaying an almost exclusive expression in this subset. Furthermore, areas of high cytokine production (TNF- α + & IFN- γ +) contain cells which either co-express KLRB1 and KLRG1 (pink arrows) or KLRG1 and GPR56 (purple arrows).

This visual inspection of the t-SNE plots indicated that different surface marker combinations are characteristic for different functional states. As the acquisition or loss of cytokine expression potential is generally linked to the differentiation state of T cells, we wanted to analyse how the expression of our surface markers correlates to the differentiation pathway of memory T cells according to the CD45RA/CCR7-based classification.

For this, we applied the recently described wanderlust algorithm to construct a trajectory of CD4⁺ T cell differentiation based on the classical (“canonical”) surface marker CD45RA and CCR7 and our identified surface marker set. Using CD45RA and CCR7 expression we defined CD45RA⁺CCR7⁺ (T_N) cells

as the “initiator” cells. This means that the classical or “canonical” differentiation path and not our newly identified marker determined the start point of the wanderlust plot.

We then examined the relative expression pattern of our identified marker but also intracellular TNF- α and IFN- γ along the developmental trajectory by plotting them against the wanderlust axis (figure 3c). According to this analysis, KLRB1 expression was the first marker to be acquired during CD4⁺ memory T cell differentiation meaning during transition from T_N to T_{CM}, a result which is in accordance with our bulk and single cell-based gene expression analyses (figures 1 and 2). Subsequently, cells started to up-regulate KLRG1 followed by a nearly parallel induction of GPR56. KLRF1 expression was only acquired at a late stage during memory T cell differentiation during the phase of CD45RA re-expression.

Interestingly, simultaneously to the up-regulation of KLRB1 T cells obtained the potential to produce TNF- α and with a slight delay also IFN- γ . Whereas KLRB1 and KLRG1 showed a nearly constant increase in expression during differentiation, GPR56 and KLRF1 expression followed a two-phase pattern. Late stage differentiated CD45RA re-expressing CD4⁺ T_{EMRA} cells acquired very high KLRG1, GPR56 and KLRF1 expression but a reduction in KLRB1 expression concurrent with a decline in TNF- α and IFN- γ production.

We hope that this more extended description is more precise but also understandable for non-experts. Furthermore, as our conventional fluorochrome-based analysis allowed us to include all necessary surface and cytokine marker we have not conducted additional CyTOF analysis for confirmation.

3) In Figure 7, the authors showed that KLRB1 single positive cells give rise to KLRB1+KLRG1+ and KLRB1+KLRG1+GPR56+. Here, they only showed the bar plots, and it looks like, if we sum up the figures, it does not reach 100. I wonder where the other fractions, if any, have gone. It would be beneficial to show usual dot plots of the FACS data to clarify what kind of populations are there, for example, to represent whether the KLRB1 negative cells may exist or not.

We agree with the reviewer that the results shown for the *in vitro* differentiation experiments of KLRB1+ single positive cells did not sum up to 100%. With those previous experiments we aimed on dissecting whether a subpopulation can further differentiate along our proposed pathway.

In order to investigate that in more detail, we improved the purity of sorted populations by setting more stringent gates and have performed additional experiments by sorting the first four subpopulations (1 = KLRB1-KLRG1-GPR56-KLRF1-, 2 = KLRB1+KLRG1-GPR56-KLRF1-, 3 = KLRB1+KLRG1+GPR56-KLRF1-, 4 = KLRB1+KLRG1+GPR56+KLRF1-) from peripheral blood of healthy individuals followed by a polyclonal *in vitro* stimulation with plate-bound anti-CD3 / CD28 antibodies. Unfortunately, the cell number of the fifth quadruple positive population (5 = KLRB1+KLRG1+GPR56+KLRF1+) was too low to perform *in vitro* stimulation experiments. We now also show the proportions of all eight defined subpopulations and the proportion of cells expressing the remaining marker combinations.

The obtained results are shown below and in figure 7 E & F of the revised manuscript.

((E) KLRB1, KLRG1, GPR56 & KLRF1 protein expression profile upon 48 h anti-CD3/CD28 mAb *in vitro* stimulation of indicated sorted CD4⁺ T cell populations from PBMCs of healthy controls (n = 5). (F) Proportions of TNF- α /IFN- γ co-producing cells of *in vitro* differentiated populations upon 96 h anti-CD3/CD28 mAb *in vitro* stimulation of indicated sorted CD4⁺ T cell populations from PBMCs of healthy controls (n = 5, mean \pm SEM). Due to the low frequency of population 4 & 5 within PBMCs, cytokine analysis was only feasible for starting populations 1, 2 and 3.

The results demonstrate, that the majority of the cells from the sorted populations keep their marker expression, meaning that e.g. only 2% of KLRB1+KLRG1-GPR56-KLRF1- cells (starting population 2) do become KLRB1-. Although we improved our cell sorting strategy to increase the purity remaining impurities may also partially explain the observed loss of marker expression.

Most importantly, the investigations also revealed that the populations further differentiate along the proposed path with e.g. KLRB1+KLRG1+GPR56-KLRF1- cells (starting population 3) acquiring also GPR56 and KLRF1 expression.

In contrast to the other populations, subset 3 (KLRB1+KLRG1+GPR56-KLRF1-) did show a higher degree of plasticity with approximately 40% of the cells becoming KLRB1+KLRG1-GPR56-KLRF1-.

In addition, we analyzed and compared the intracellular TNF- α & IFN- γ expression of the *in vitro* differentiated subsets (see figure 7F within the revised manuscript). Due to cell number limitations upon FACS-based enrichment the experiments could be only done with the following starting populations: 1 = KLRB1-KLRG1-GPR56-KLRF1-, 2 = KLRB1+KLRG1-GPR56-KLRF1-, 3 = KLRB1+KLRG1+GPR56-KLRF1-.

The results show that also upon *in vitro* differentiation the subpopulations display the expected evolution in proportions of TNF- α & IFN- γ double producers. The highest frequency was observed for KLRB1+KLRG1+GPR56-KLRF1- & KLRB1+KLRG1+GPR56+KLRF1- cells, whereas a decline was observed upon acquisition of KLRF1 expression.

We hope by performing these additional experiments and showing the proportions of all populations we could clarify this issue. If it is still necessary to show all original dot plots of *in vitro* differentiated populations (20 per experiment) we are happy to provide those data as supplementary figures.

4) Related to the previous point, the authors examined the expression patterns of the limited combination (namely KLRB1 vs KLRG1 and KLRF1 vs GPR56) of the four markers in Figure 2. It would be informative to show other patterns to see if any other populations than the proposed path (in Figure 4) exist. The transcriptome-wide analysis as described above would be even better.

In figure 2 we show the change in expression for each of the four markers during CD4⁺ memory T cell differentiation from T_N via T_{CM} and T_{EM} towards T_{EMRA}. Thus, within this figure we did not aim on defining subsets or populations based on the surface marker expression but rather to reveal differences in total marker expression during CD4⁺ memory T cell differentiation. We displayed the expression as KLRB-1

versus KLRG-1 and GPR56 versus KLRF1 as the single cell gene expression analysis revealed that *KLRB1* transcription was observed in T_{CM} and T_{EM} cells, and *KLRG1* transcription was detectable in T_{EM} and T_{EMRA} cells whereas *GPR56* and especially *KLRF1* transcription was only detectable in T_{EMRA} cells.

To clarify this issue, we have now modified the corresponding results section as follows:

We detected a successive increase in expression of all markers starting from T_N to T_{EMRA} cells (figure 2a) with *KLRB1* being already up-regulated at an early memory stage (T_{CM} cells). In contrast, expression of the other three marker increased later during memory / effector cell development. *KLRG1* was expressed by approximately 50% of the T_{EM} and nearly all T_{EMRA} cells. *GPR56* and especially *KLRF1* up-regulation occurred even later during differentiation with only T_{EMRA} cells displaying a relatively high expression.

Subset definition based on marker combinations was only done after t-SNE and Wanderlust analysis, which indicated progressive accumulation of marker expression starting with *KLRB1* and subsequent induction of *KLRG1*, *GPR56* and finally *KLRF1*.

Furthermore, the subsets defined based on our proposed marker combinations make up nearly 100% of T_{EM} and T_{EMRA} cells (figure 4C) indicating that other populations than the proposed path constitute a rather small fraction.

5) Further related to the previous points, if the remaining approximately 70 % of the cells, as presented in Figure 7, had already lost the *KLRB1* expression, this would lead to a serious question on the robustness of the proposed classification scheme. To what extent are the proposed populations, such *KLRB1+*, *KLRB1+KLRG1+* and so on, likely to have certain plasticity? In other words, the robustness of the proposed path as shown in Figure 4 should be further evaluated. For example, if the cells in a population such as *KLRB1+KLRG1+GPR56+KLRF1-* are sorted and stimulated, can we expect if all of the cells would turn out to be “quadruple-positive” population. Alternatively, would some cells be branched off from the path?

We thank the reviewer for raising this important point. As already mentioned in the reply to comment 3 we have performed additional *in vitro* stimulation experiments of all FACS-enriched populations. The results show, that the majority of the cells from the sorted populations keep their marker expression, meaning that e.g. only 2% of *KLRB1+KLRG1-GPR56-KLRF1-* cells (starting population 2) do become *KLRB1-* or that 20% of the *KLRB1+KLRG1+GPR56+KLRF1-* cells (starting population 4) lose *GPR56* expression.

Impurities after the FACS-based enrichment may in part explain the observed loss of marker expression.

Interestingly, in contrast to the other populations, subset 3 (*KLRB1+KLRG1+GPR56-KLRF1-*) did show a higher degree of plasticity with approximately 40% of the cells becoming *KLRB1+KLRG1-GPR56-KLRF1*. Most importantly, the investigations also revealed that the populations further differentiate along the proposed path with e.g. *KLRB1+KLRG1+GPR56-KLRF1-* cells (starting population 3) acquiring also *GPR56* and *KLRF1* expression.

In summary, these data provide further evidence that a) the expression of these four differentiation markers on isolated CD4+ T cell subsets in general is robust, with the *KLRB1+KLRG1+* double positive subset showing a higher degree of plasticity, and b) that T cells can express additional markers upon re-stimulation supporting the validity of the proposed differentiation pathway.

6) The authors discussed the PD-1 expression in the canonical categorization (in Discussion section).

What about the expression of the other markers for exhaustion such as TIM-3, LAG-3 and so on in the proposed populations?

We agree with the reviewer that analysis of other well-known T cell coinhibitory / exhaustion markers is an important aspect. We already showed the expression of TIGIT within the canonical subsets, but have now also included the expression analysis for TIM-3 and LAG-3 (Supplementary figure 1A within the revised manuscript). The results show that expression of TIM-3 is not limited to one subset and is even highest in CD4⁺ T_N cells. Similarly, LAG-3 expression was high in T_N and T_{CM} cells and low in T_{EM} and T_{EMRA} cells. In comparison, we now also show the expression in our newly proposed KLR/GPR56-based subsets (Supplementary figure 1B). Interestingly, TIGIT and PD-1 show a variable expression during the course of our proposed differentiation pathway. We observed a medium expression level in the first subset

(1 = KLRB1-KLRG1-GPR56-KLRF1-), an increase in expression was observed for the second subset (2 = KLRB1+KLRG1-GPR56-KLRF1-). Thereafter, in the high cytokine expressing subsets (3 = KLRB1+KLRG1+GPR56-KLRF1- & 4 = KLRB1+KLRG1+GPR56+KLRF1-) the expression declined followed by a dramatic increase in the final fifth subset (5 = KLRB1+KLRG1+GPR56+KLRF1+). In contrast, TIM-3 and LAG-3 expression was only observed for the first two subsets and down-regulated upon further differentiation.

7) Indeed, I could not clearly see how the results should be compared between this study in humans and previous studies mainly in mice. Providing a concise table in supplementary information with this regard would be extremely helpful.

We thank the reviewer for this helpful advice. In the revised manuscript we have now included a table (Supplementary table 4) summarizing the expression pattern of previously defined and our proposed surface markers on human and murine CD4⁺ and CD8⁺ T cells to distinguish between activated, memory and dysfunctional / exhausted states.

8) I sincerely hope the authors further pursue the path to elucidate the molecular mechanisms of the gene expression changes, also in conjunction with other previous knowledge. I believe such study should be no less important than to investigate the clinical relevance of the new findings described here. After all, we would not be able to totally rely on the phenomenon for which the mode-of-action still remains elusive.

We totally agree with the reviewer that further studies on the molecular mechanisms that regulate the expression of the here investigated markers during T cell differentiation and the relevance of the described T cell phenotypes for different clinical conditions are warranted.

We believe that the analysis of TCR clonotypes by sequencing of TCR beta chains represents an important first step to provide further evidence for the molecular relationship and differentiation status of our proposed populations. Therefore, we performed FACS-based enrichment of all five populations (1 = KLRB1-KLRG1-GPR56-KLRF1-, 2 = KLRB1+KLRG1-GPR56-KLRF1-, 3 = KLRB1+KLRG1+GPR56-KLRF1-, 4 = KLRB1+KLRG1+GPR56+KLRF1-, 5 = KLRB1+KLRG1+GPR56+KLRF1+) from peripheral blood of healthy individuals followed by a TCR beta chain sequencing analysis.

Interestingly, the TCR beta chain sequencing analysis revealed that the clonal frequency of T cells belonging to the KLRB1-KLRG1-GPR56-KLRF1- subpopulation was rather low, whereas a progressive increase in clonal frequency was observed for T cells belonging to the other populations (see figure below and figure 7A within the revised manuscript).

Subsequently, the TCR diversity decreased along our proposed differentiation path (see figure below and figure 7B within the revised manuscript).

Furthermore, the proposed subpopulations do not differentiate completely independent from each other as TCR profiles overlap especially between the late populations 4 & 5 but also 3 (see figure below and figure 7C within the revised manuscript) and TCR clones dominating in the late populations 4 & 5 can be even found in the early populations 1 & 2 (see figure below and figure 7D within the revised manuscript).

Clonal space (A), Rényi diversity profile (B, two exemplary), clonal similarity (C, two exemplary) of TCR β chain of sorted KLR/GPR56⁻ (population 1), KLRB1⁺ (population 2), KLRB1⁺KLRG1⁺ (population 3), KLRB1⁺KLRG1⁺GPR56⁺ (population 4) and KLRB1⁺KLRG1⁺GPR56⁺KLRF1⁺ (population 5) CD4⁺ T cells from healthy controls (n=5). (D) Number of TCR clonotypes dominating in source population 4 & 5 identified in target populations 1, 2 & 3. The scaling factor Alpha of the Rényi diversity profile yields the sample diversity with different weighting of the clonotype proportion, see Materials and Methods. The clonotypes were verified prior to diversity calculation. The clonal similarity was assessed using the index of Morisita-Horn (1 indicates identity). Data are accessible within the European Nucleotide Archive (ENA) Accession Number ,PRJEB31283'.

These results further provide evidence for a linear differentiation along and molecular relationships between the proposed populations.

Future investigations may provide additional information on the role of these surface markers for cytokine regulation, but this is clearly beyond the scope of the current manuscript.

Reviewer #2 (Memory T cell biology) (Remarks to the Author):

In this manuscript Truong et. al. put forth an alternative classification scheme for CD4⁺ memory T cells

based on surface markers related to cytokine production instead of the classical TCM, TEM, TEMRA divisions. They use cytokine production to define surface markers within the TEM and TEMRA populations relevant to 'high producers', or conversely, 'exhausted' cells. They observe KLRF1 to be the most relevant marker to identify exhausted CD4 T cells whereas KLRB1 is associated with high cytokine production. Additionally, they assign a differentiation scheme in which acquisition of NK-associated markers KLRB1, KLRG1, GPR56, and KLRF1 occurs in a successive and linear fashion leading to first improved, then declining function. There are several significant issues with the manuscript which need to be addressed before it should be considered for publication. Specific points are outlined below.

Specific points:

1. What is the rationale for combining TEM and TEMRA in Figure 1B when most of the paper keeps these two as separate cell groups? What is the scale for the heat map in Figure 1B?

We like to thank the reviewer for pointing out that the rationale for combining the transcriptomics results for the intersection analysis of T_{EM} / T_{EMRA} versus T_N / T_{CM} cells did not become clear in our previous version of the manuscript.

Performing two separate cluster analyses allowed us to identify genes that were highest in T_{EMRA} cells (T_{EMRA} vs. T_N / T_{CM} , T_{EM} cells) but also already up-regulated in T_{EM} cells (T_{EMRA} / T_{EM} vs. T_N / T_{CM} cells) in comparison to earlier T_N and T_{CM} differentiation stages.

However, we noticed a mix-up in the figure legend and corresponding results section. Figure 1A shows the cluster analysis for the comparison of T_{EMRA} / T_{EM} vs. T_N / T_{CM} cells, whereas figure 1B shows the intersection results for the comparison of T_{EMRA} vs. T_N / T_{CM} , T_{EM} cells. We apologise for the mistake which we have corrected in the revised manuscript.

The scale for Figure 1B is identical to the one shown in Figure 1A. This is now described in the figure legend.

2. (Figure 1C, D); 200-400 cells were analyzed by qRT-PCR, followed by a cut off of 10 cells expressing a gene to then perform clustering analysis. 4 donors were used in total. This is a very low number of samples and cells to score as a significant finding, particularly given that strong heterogeneity in the TEMRA pool between donors has been noted before.

We have repeated the single cell qRT-PCR analysis of sorted T_{EM} and T_{EMRA} cells for an additional healthy donor and now present results from five different donors (see figure below & new Supplementary figure 1 within revised manuscript).

The unsupervised cluster analysis of this new data set has resulted in a nearly identical expression pattern. Thus, we could validate the relative homogenous KLRG1 and heterogenous KLRF1 expression in T_{EMRA} cells. The combined analysis is part of the revised manuscript.

Therefore, although the frequency of $CD4^+$ T_{EM} and T_{EMRA} cells does vary between individual healthy donors, their composition and homogenous versus heterogenous expression pattern of our identified marker appears to be stable.

(A) Previous unsupervised cluster analysis of single cell gene expression results from identified NK cell-associated markers in blood CD4⁺ T_{EM} (199) and T_{EMRA} (226) cells of 4 healthy individuals revealing homogeneous (e.g. KLRG1) or heterogeneous (e.g. KLRF1) expression pattern.

(B) Unsupervised cluster analysis of single cell gene expression results from identified NK cell-associated markers in blood CD4⁺ T_{EM} (n = 61) and T_{EMRA} cells (n = 50) **from one additional** healthy individual validating homogeneous (e.g. KLRG1) or heterogeneous (e.g. KLRF1) expression pattern in CD4⁺ T_{EMRA} cells.

(C) Unsupervised hierarchical cluster analysis of the combined single cell gene expression results from identified natural killer cell-associated markers in blood CD4⁺ T_{EM} (n = 260) and T_{EMRA} (n = 276) cells of five healthy individuals. (Figure 1D in the revised manuscript)

Classification as expressing and non-expressing cells based on individual defined limit of detection (LoD) Ct values. Data are provided with the source data file.

3. An issue with this manuscript is that it's sometimes hard to follow. There are so many permutations of the data displayed that the main point(s) get lost. I would suggest limiting the data display (or moving more to supplemental) to make the main points more evident. For example, in Figure 3, remove IL-4 and IL-17A as cells display minimal to no production. Figure 5 is also hard to digest in its current form.

We thank the reviewer for this helpful advice. As suggested by the reviewer we have moved the dot plots and bar graphs showing the IL-4 and IL-17A expression to the new Supplementary figure 2. Furthermore, we have moved the previous figure 5C showing the frequencies of T_{EM} and T_{EMRA} cells transcribing the gene marker to the new Supplementary figure 4 and additionally have removed the bar graphs for T_{CM} cells.

4. The evidence to indicate the linear progression of cells from KLRB1+ to KLRB1+KLRG1+ and so on is not convincing. There is a lot of emphasis on Figure 3C for this claim (Wanderlust analysis), but that makes assumptions on linear differentiation from the onset. The authors then assume this analysis is correct and draw arrows in Figure 4 to indicate how the differentiation pathway relates to cytokine production.

Indeed, further evidence for linear progression of our populations would strengthen our hypothesis. We believe that the analysis of TCR clonotypes by sequencing of TCR beta chains represents an important to provide such evidence for a) molecular relationship and b) differentiation pathway of our proposed populations.

Thus, to provide evidence for a linear progression from quadruple negative (KLRB1-KLRG1-GPR56-KLRF1-) to single (KLRB1+), double (KLRB1+ KLRG1+) and finally quadruple positive (KLRB1+KLRG1+GPR56+KLRF1+) CD4⁺ T cells we have performed TCR beta chain sequencing analyses of FACS-sorted cell populations (please also see reply to comment 8 of reviewer 1).

Interestingly, the TCR beta chain sequencing analysis revealed that the clonal frequency of T cells belonging to the KLRB1-KLRG1-GPR56-KLRF1- subpopulation was rather low, whereas a progressive increase in clonal frequency was observed for T cells belonging to the other populations (see figure 7A within the revised manuscript).

Subsequently, the TCR diversity decreased along our proposed differentiation path (see figure 7B within the revised manuscript).

Furthermore, the proposed subpopulations do not differentiate completely independent from each other as TCR profiles overlap especially between the late populations 4 & 5 but also 3 (see figure 7C within the revised manuscript) and TCR clones dominating in the late populations 4 & 5 can be even found in the early populations 1 & 2 (see figure 7D within the revised manuscript).

In addition, we have extended our *in vitro* differentiation experiments of FACS-enriched populations and show that the populations further differentiate along the proposed path with e.g. KLRB1+KLRG1+GPR56-KLRF1- cells (starting population 3) acquiring also GPR56 and KLRF1 expression (see figure 7E within the revised manuscript).

These results clearly provide further evidence for a linear differentiation along and molecular relationships between the proposed populations.

5. Figure 7 provides some evidence to support their differentiation theory, but it falls short of being convincing. Where are the other 80% of cells in this assay? What happens if you start with KLRB1+KLRG1+ cells, KLRF1-expressing cells, etc.? This is a critical experiment to support the underlying differentiation conclusion which is central to the paper. In 7B, there is no negative control to compare the cytokine production to, nor are single positive KLRB1-expressing cells included.

We agree with the reviewer that the results shown for the *in vitro* differentiation experiments of KLRB1+ single positive cells did not sum up to 100%. With those previous experiments we aimed on dissecting whether a subpopulation can further differentiate along our proposed pathway.

In order to investigate that in more detail, we improved the purity of sorted populations by setting more stringent gates and have performed additional experiments by sorting the first four subpopulations (1 = KLRB1-KLRG1-GPR56-KLRF1-, 2 = KLRB1+KLRG1-GPR56-KLRF1-, 3 = KLRB1+KLRG1+GPR56-KLRF1-, 4 = KLRB1+KLRG1+GPR56+KLRF1-) from peripheral blood of healthy individuals followed by a polyclonal *in vitro* stimulation with plate-bound anti-CD3 / CD28 antibodies (see also reply to comment 3 of reviewer 1).

Unfortunately, the cell number of the fifth quadruple positive population (5 = KLRB1+KLRG1+GPR56+KLRF1+) was too low to perform *in vitro* stimulation experiments. We now also show the proportions of all eight defined subpopulations and the proportion of cells expressing the remaining marker combinations, which sums up to 100%.

The obtained results are shown in figure 7 E & F of the revised manuscript.

The results show, that the majority of the cells from the sorted populations keep their marker expression, meaning that e.g. only 2% of KLRB1+KLRG1-GPR56-KLRF1- cells (starting population 2) do become KLRB1- or that 20% of KLRB1+KLRG1+GPR56+KLRF1- cells (starting population 4) lose GPR56 expression.

Although we improved our cell sorting strategy to increase the purity remaining impurities may also partially explain the observed loss of marker expression.

In contrast to the other populations, subset 3 (KLRB1+KLRG1+GPR56-KLRF1-) did show a higher degree of plasticity with approximately 40% of the cells becoming KLRB1+KLRG1-GPR56-KLRF1-.

Most importantly, the investigations also revealed that the populations further differentiate along the proposed path with e.g. KLRB1+KLRG1+GPR56-KLRF1⁻ cells (starting population 3) acquiring also GPR56 and KLRF1 expression.

In addition, we analyzed and compared the intracellular TNF- α & IFN- γ expression of the *in vitro* differentiated subsets (see figure 7F within the revised manuscript). Due to cell number limitations upon FACS-based enrichment the experiments could be only done with the following starting populations: 1 = KLRB1-KLRG1-GPR56-KLRF1⁻, 2 = KLRB1+KLRG1-GPR56-KLRF1⁻, 3 = KLRB1+KLRG1+GPR56-KLRF1⁻.

However, the results show that also upon *in vitro* differentiation the subpopulations display the expected evolution in proportions of TNF- α & IFN- γ double producers. The highest frequency was observed for KLRB1+KLRG1+GPR56-KLRF1⁻ & KLRB1+KLRG1+GPR56+KLRF1⁻ cells, whereas a decline was observed upon acquisition of KLRF1 expression.

6. Throughout the manuscript the authors indicate Cxc3cr1 cells. Is this supposed to be CX3CR1?

We apologize for the typing errors, which we have corrected it within the revised manuscript.

REVIEWERS' COMMENTS:

Reviewer #1 (Remarks to the Author):

First of all, I appreciate the dedicated efforts of the authors made for the revision. With the substantial amount of the results from the extensive analyses and the deepened discussion, I believe the paper is now thorough and convincing. Indeed, it has been my own concern that the memory T cell could not be clearly separated into functionally relevant groups solely based on the current criteria. Based on the presented solid biological evidence, I believe this paper should bring practically useful information to other researchers having the same problem. Of course, different issues would rise, regarding the further in-depth characterization of the respective sub-cell types, but I would consider it should be the subject of future study.

Reviewer #2 (Remarks to the Author):

Truong, et.al. have made several worthwhile changes to the manuscript and have satisfied the criticisms raised during first review. In particular, the changes to Figure 7 were a nice addition. I do not have any remaining concerns.